# Adversarial Vulnerability from On-Manifold Inseparability and Poor Off-Manifold Convergence

**Rajdeep Haldar** *rhaldar@purdue.edu*
*Department of Statistics, Purdue University*

**Yue Xing** *xingyue1@msu.edu*
*Department of Statistics and Probability, Michigan State University*

**Qifan Song** *qfsong@purdue.edu*
*Department of Statistics, Purdue University*

**Guang Ling** *guanglin@purdue.edu*
*Department of Statistics, Purdue University*

**Reviewed on OpenReview:** *https://openreview.net/forum?id=pa9OuRZATF*

## Abstract

We introduce a new perspective on adversarial vulnerability in image classification: fragility can arise from poor convergence in off-manifold directions. We model data as lying on low-dimensional manifolds, where on-manifold directions correspond to high-variance, data-aligned features and off-manifold directions capture low-variance, nuanced features. Standard first-order optimizers, such as gradient descent, are inherently ill-conditioned, leading to slow or incomplete convergence in off-manifold directions. When data is inseparable along the on-manifold direction, robustness depends on learning these subtle off-manifold features, and failure to converge leaves models exposed to adversarial perturbations.

On the theoretical side, we formalize this mechanism through convergence analyses of logistic regression and two-layer linear networks under first-order methods. These results highlight how ill-conditioning slows or prevents convergence in off-manifold directions, thereby motivating the use of second-order methods which mitigate ill-conditioning and achieve convergence across all directions. Empirically, we demonstrate that even without adversarial training, robustness improves significantly with extended training or second-order optimization, underscoring convergence as a central factor.

As an auxiliary empirical finding, we observe that batch normalization suppresses these robustness gains, consistent with its implicit bias toward uniform-margin rather than max-margin solutions.

By introducing the notions of on- and off-manifold convergence, this work provides a novel theoretical explanation for adversarial vulnerability.

## 1 Introduction

Neural networks achieve high classification accuracy and generalize well to unseen examples drawn from the training distribution (Liu et al., 2020). Yet, they exhibit a striking vulnerability: small, imperceptible perturbations can drastically alter predictions (Szegedy et al., 2013; Goodfellow et al., 2014; Madry et al., 2017; Carlini & Wagner, 2017). These perturbed inputs, known as *adversarial examples*, highlight fundamental flaws in modern training and raise questions about the reliability of neural networks in safety-critical applications.

While empirical progress has been rapid, yielding strong attacks and defenses, the theoretical foundations for why adversarial examples exist have advanced more slowly. A principled understanding is critical for developing models that are inherently robust and aligned with human perception.

**The manifold hypothesis.** One of the most compelling explanations is the *manifold hypothesis*, which posits that real-world data lies on a low-dimensional manifold embedded in a high-dimensional space. Recent works (Melamed et al., 2024; Haldar et al., 2024) show that redundant dimensions (e.g., background pixels in images) make it possible to generate adversarial perturbations of arbitrarily small magnitude. This suggests that robustness may be attainable if learning is confined to the manifold.

**Implicit bias and robustness expectations.** The implicit bias literature (Wei et al., 2019; Lyu & Li, 2019; Lyu et al., 2021; Nacson et al., 2022) demonstrates that standard training often yields solutions corresponding to KarushKuhnTucker (KKT) points of the *maximum-margin problem*. More generally, Rosset et al. (2003) established that losses like logistic regression are inherently margin-maximizing, a property extending to deep networks. These results suggest that in the absence of redundant dimensions, clean training is expected to yield max-margin classifiers that possess reasonable robustness.

**The puzzle.** In practice, however, this expectation is not realized. Dimensionality reduction methods such as PCA do not lead to the anticipated robustness gains (Alemany & Pissinou, 2020; Aparne et al., 2022), while adding redundant dimensions consistently amplifies vulnerability. This raises fundamental questions: Is clean training alone sufficient for robustness? And are there overlooked mechanisms that undermine the robustness promised by implicit bias?

**Our perspective.** In this work, we identify such a mechanism. We argue that *poor convergence in off-manifold directions*, caused by ill-conditioning of first-order optimization, is an additional and overlooked source of adversarial vulnerability. Our theoretical framework is summarized in Figure 1. High-variance features define *on-manifold* directions, while low-variance features define *off-manifold* directions. First-order methods converge quickly in on-manifold directions but slowly off-manifold, leading to suboptimal boundaries when separability relies on off-manifold features. This convergence gap explains why robustness does not follow automatically from implicit bias, even in the absence of redundant dimensions.

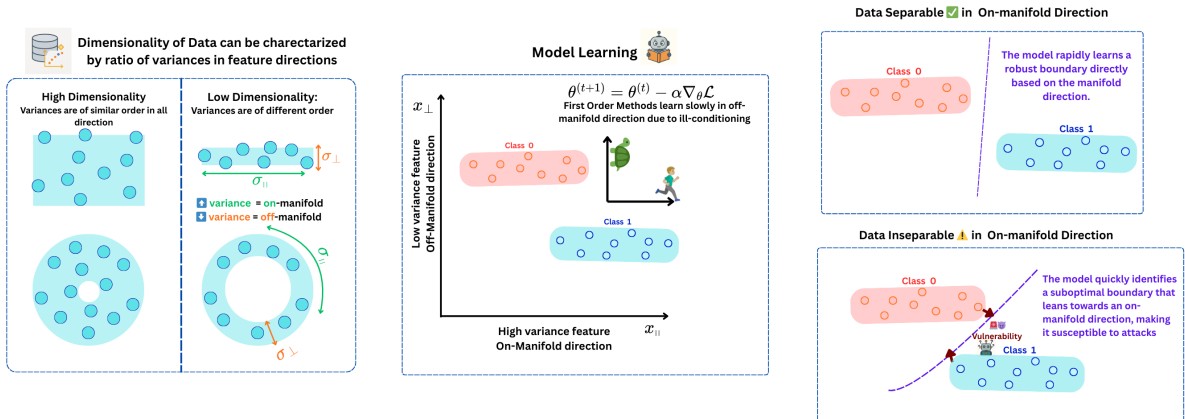

Figure 1: Theoretical framework for adversarial vulnerability from a convergence perspective. (Left) Data dimensionality can be characterized by the ratio of variances in feature directions: high-variance features define on-manifold directions, while low-variance features define off-manifold directions. (Middle) First-order methods suffer from ill-conditioning in off-manifold directions, leading to slow convergence. (Right) When data is separable along the on-manifold direction, models quickly learn robust boundaries. When inseparable, however, convergence failure in off-manifold directions causes suboptimal boundaries and increased adversarial vulnerability.

**Mathematical intuition for dimensionality in terms of variance.** We view variance ratios across feature directions as a surrogate for the effective dimensionality of the data manifold. Intuitively, directions with high variance capture the dominant structure of the data (on-manifold), while directions with very low variance contribute little and can be regarded as off-manifold. As the variance in certain directions vanishes relative to others, the data distribution effectively collapses to a lower-dimensional set. This provides a natural way to formalize dimensionality beyond earlier notions based only on redundant versus useful features (Haldar et al., 2024; Melamed et al., 2024). Our definition extends these ideas by characterizing on- and off-manifold directions explicitly through variance ratios.

**Illustrative examples.** Figure 2 provides intuition for our framework. In a birdinsect classification task (Fig. 2a), wing length serves as a high-variance, on-manifold feature that separates most birds from insects. A classifier trained on this feature alone will perform well on the majority of examples, and thus naturally favors it. However, overlap occurs: certain insects, such as moths, may have wings longer than some small birds, such as hummingbirds. In this overlapped region, robustness requires exploiting a low-variance, off-manifold featuresuch as the presence of a beak. Because off-manifold features are harder to learn due to slower convergence, models that fail to capture them remain fragile.

This phenomenon is demonstrated in a synthetic experiment (Fig. 2b). When the data is separable along the on-manifold direction, neural networks rapidly learn stable, robust decision boundaries. In contrast, when the data is inseparable on-manifold, distinguishing classes requires reliance on off-manifold directions. Here, convergence is markedly slower, and the learned boundaries remain suboptimal for many training epochs despite high classification accuracy. Together, these examples highlight why models often default to on-manifold features, yet robustness in overlapping regions critically depends on off-manifold learning, which is precisely where first-order methods struggle.

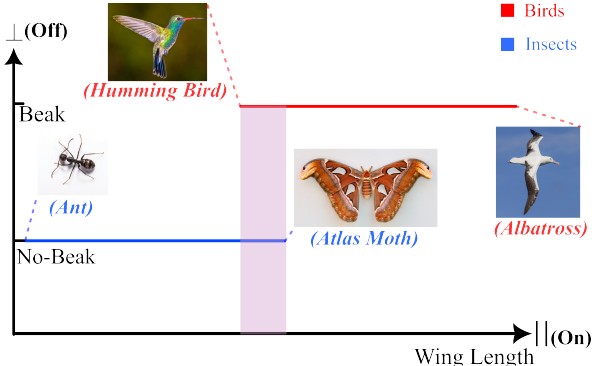

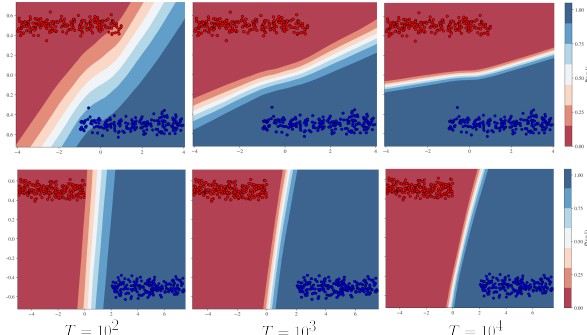

(a) Real-world example. Binary classification between birds and insects. The purple region shows overlap in the on-manifold feature, where only the off-manifold feature (beak vs. no-beak) separates the classes.

(b) Synthetic example. Neural network decision boundaries after $T$ training epochs. Data distribution is (top) inseparable on-manifold and (bottom) separable. Clean accuracy is 100% for all $T \geq 10^2$, yet convergence is fast and slow across on- and off-manifold directions respectively, which may lead to suboptimal decision boundary.

Figure 2: Illustrations of adversarial vulnerability arising from off-manifold convergence. (a) A real-world case where overlap in the on-manifold feature requires reliance on off-manifold features. (b) A synthetic example showing how first-order methods converge differently along on- vs. off-manifold directions.

**Our contributions.** We formalize the ideas described in Figs. 1 and 2 through parameter and loss convergence results:

**Theorem 1.1** (Informal version of Theorem 4.2)**.** *Convergence is faster and independent of dimensionality in on-manifold directions, but slower in off-manifold directions. As dimensionality reduces, off-manifold convergence becomes even slower.*

**Theorem 1.2** (Informal version of Theorem 4.3). *There exists an error threshold, determined by on-manifold separability, that is quickly reached. Reducing error further depends on off-/on-manifold variance ratios and is substantially slower.*

**Implications.** These results clarify why clean training may give the *illusion* of convergence: models can achieve high training accuracy after finite iterations, yet incomplete convergence across all directions leaves them vulnerable to adversarial perturbations. Addressing convergence in off-manifold directions leads to *substantial robustness improvements*, even without adversarial training. While this does not eliminate adversarial vulnerability entirely, the gains are significant. Furthermore, second-order optimization methods, which are immune to ill-conditioning, realize these improvements far more efficiently (Section 5), a finding validated by our experiments (Section 6.2).

## 2 Related Works

**Adversarial Vulnerability Explained via Manifold Hypothesis:** Stutz et al. (2019) introduces the distinction between *on/off-manifold adversarial examples* and their role in robustness and generalization. Shamir et al. (2021) shows empirically that clean-trained classifiers align with a low-dimensional *dimple-manifold* around the data, making them vulnerable to imperceptible perturbations. Zhang et al. (2022) decomposes the adversarial risk into geometric components motivated by manifold structure. Haldar et al. (2024); Melamed et al. (2024) theoretically establish a relationship between the *dimension gap* (induced by low-dimension data manifold) and adversarial vulnerability.

**Attack/Defense systems based on manifold hypothesis:** Xiao et al. (2022) uses a generative model to learn the manifold and create on-manifold attacks. Jha et al. (2018); Lindqvist et al. (2018); Lin et al. (2020) have tried to learn the underlying manifold to detect and defend adversarial examples outside the data manifold.

**Implicit Bias and Robustness** Frei et al. (2024) shows that the max-margin implicit bias of neural networks isn't the most robust model. However, the attack strength in this work is comparable to the signal strength, and reduces to natural/on-manifold attack scenario mentioned in Haldar et al. (2024). These attacks are perceptible by humans and of relatively large magnitude. We still expect natural robustness against imperceptible or small-magnitude attacks in the absence of redundant dimensions, as discussed previously. Min & Vidal (2024) shows that under poly-RelU activation, implicit bias is robust to larger attacks.

**Optimization Dynamics and feature learning** A growing body of work studies how training dynamics influence robustness. Kalimeris et al. (2019) showes that SGD learns low-complexity functions before more complex ones, while Ilyas et al. (2019), Hermann et al. argue that adversarial vulnerability arises due to shortcut learning of more accessible non-robust features. Tsilivis & Kempe (2022) analyzes adversarial robustness through the NTK spectrum, showing that robust features correspond to dominant eigen-directions learned early in training. Rice et al. (2020) further demonstrates that adversarial training can overfit due to optimization dynamics. Tanay & Griffin (2016) attribute adversarial examples to decision boundaries lying close to the data manifold. Our framework offers a complementary explanation via convergence: even when robust features are linearly separable and accessible, first-order methods converge slowly along low-variance (off-manifold) directions due to ill-conditioning, thereby delaying robust solutions when the data isn't separable in the fast-converging manifold direction. Additionally, Tanner et al. (2025) and Javanmard & Soltanolkotabi (2022) also provide data geometric arguments in studying robustness for adversarial training. In contrast, our work strictly focuses on standard training and how the lack of convergence due to data geometry can inherently limit robustness, despite the existence of a robust solution.

To the best of our knowledge, our work is the first to extend the notion of on/off manifold dimensions in terms of feature variances rather than utility, and to draw connections between on-manifold separability, ill-conditioning, and adversarial vulnerability.

# 3 Problem Setup

This section describes the technical setting and assumptions used for our main results in section 4.2.

## 3.1 Notation

Throughout this paper, we will use the subscripts $\shortparallel, \perp$ to denote mathematical objects corresponding to on-manifold and off-manifold, respectively. The notation $\mathbb{1}_d$ denotes the concatenated vector of ones of length $d$. For any two vectors $u, v$ such that $u_i \leq v_i \ \forall i$, we can define a hypercube $[u, v]$ such that if $z \in [u, v]$ then for each $i$, $z_i \in [u_i, v_i]$. Denote $\mathbf{I}_d$ as the identity matrix of $d$ dimensions. Notation $\Omega(\cdot)$ is the usual asymptotic lower bound notation. For $a, b \in \mathbb{R}$, $a \preceq b \iff a \leq b \cdot c$ for some $c > 0$. Note that the notation $\preceq$ also works for negative sequences $a$ and $b$ (i.e., $a \preceq b$ implies $|b| \preceq |a|$ when $a$ and $b$ are negative). Denote $\odot$ as the Hadamard product.

## 3.2 Assumption on Data Distribution

We denote the underlying signal of the data with $x \in \mathbb{R}^D$ and its corresponding label as $y \in \{-1, +1\}$. We study a binary classification problem with the data pair $(x, y)$.

Our signal can be decomposed as $x = (x_\shortparallel, x_\perp)$, where $x_\shortparallel \in \mathbb{R}^d$ and $x_\perp \in \mathbb{R}^g$ are the on/off manifold components respectively. Corresponding to each class, define $d$-dimensional hypercubes of side $l$ as $\mathcal{I}_\shortparallel^{(-1)} = [-(l-k) \cdot \mathbb{1}_d, k \cdot \mathbb{1}_d]$ and $\mathcal{I}_\shortparallel^{(+1)} = [-k \cdot \mathbb{1}_d, (l-k) \cdot \mathbb{1}_d]$. The parameter $k$ controls the overlap between the two hyper-cubes in the on-manifold direction. We assume that conditioned on the label the on-manifold signal is uniformly drawn from such hyper-cubes $x_\shortparallel | y \sim \mathcal{U}(\mathcal{I}_\shortparallel^{(y)})$ with means $\mu_\shortparallel^{(y)} = \frac{y(l-2k)}{2} \cdot \mathbb{1}_d$ and covariance $\sigma_\shortparallel^2 \cdot \mathbf{I}_d = \frac{l^2}{12} \cdot \mathbf{I}_d$ respectively. Similarly, define non-overlapping symmetric hyper-cubes $\mathcal{I}_\perp^{(y)} = [\mu_\perp^{(y)} - \sqrt{3}\sigma_\perp \cdot \mathbb{1}_g, \mu_\perp^{(y)} + \sqrt{3}\sigma_\perp \cdot \mathbb{1}_g]$ such that $\mathcal{I}_\perp^{(-1)} \cap \mathcal{I}_\perp^{(+1)} = \emptyset$, and $x_\perp | y \sim \mathcal{U}(\mathcal{I}_\perp^{(y)})$ with means $\mu_\perp^{(y)}$ and covariance $\sigma_\perp^2 \cdot \mathbf{I}_g$. The class probabilities themselves are binomial with probability $\pi$ i.e. $P(y = 1) = \pi$ and $P(y = -1) = 1 - \pi$. Also, $\|\mu_\perp^{(y)}\| < \infty$ and $\sigma_\perp / \sigma_\shortparallel < 1$.

To explain the above assumption, the data is essentially uniformly distributed over two high-dimensional rectangles corresponding to each class. Due to the non-overlapping off-manifold distributions, the two rectangles are linearly separable in the ambient $D$-dimensional space. However, there is an overlap in the on-manifold distribution controlled by $k$. The $x$ and $x_\perp$ distributions are linearly separable, but $x_\shortparallel$ is not for $k \neq 0$. The ratio of the variances $\sigma_\perp^2 / \sigma_\shortparallel^2 < 1$ characterizes the low-dimensional manifold structure. Our data model is a high dimensional mathematical representation of Figures 1 (middle) and 2b.

**Overlapping coefficient** In order to formulate the extent of non-separability in the on-manifold distribution we borrow the concept of *overlapping coefficient* (OVL) from traditional statistics. For any two probability densities $f_1(x), f_2(x)$ the overlapping coefficient is defined as $\nu = \int_\mathcal{D} \min(f_1(x), f_2(x)) \, dx$ where $\mathcal{D}$ is the support. Note that $\nu \in [0, 1]$, and essentially represents the probability of $x$ being drawn from the minority distribution. In the context of a naive Bayes classifier, $\nu$ represents the probability of misclassification or area of conflict. We can quantify non-separability for the on-manifold distribution by computing $\nu = \int \min\left(\mathcal{U}(\mathcal{I}_\shortparallel^{(-1)}), \mathcal{U}(\mathcal{I}_\shortparallel^{(+1)})\right) dx = (k/l)^d$. As $k$ increases, geometrically, our on-manifold hypercubes overlap more, which is consistent with $\nu$. Later, we will see how the non-separability of the on-manifold component affects our classifier learning.

## 3.3 Models

For our binary classification problem, we work with the logistic loss $\ell(z) = \ln(1 + e^{-z})$. We will denote $f(x, \gamma)$ with real output as the score predicted by our classifier for a particular data point $x$ and parameter vector $\gamma$. The expected loss of the classifier over the data distribution is $\mathcal{L}(\gamma) = \mathop{\mathbb{E}}_{\sim x, y} \ell(y \cdot f(x, \gamma))$. The clean training is the optimization problem $\min_{\gamma \in \Gamma} \mathcal{L}(\gamma)$ where $\Gamma$ is a compact parameter space. Let $\gamma^* = \arg\min_{\gamma \in \Gamma} \mathcal{L}(\gamma)$, then

working with compact space ensures $\|\gamma^*\|$ is finite and makes analysis tractable. This work considers the linear model and the two-layer linear network.

### 3.3.1 Logistic Regression

In the linear setup, $\gamma = \theta$ where $\theta \in \mathbb{R}^D$ is the coefficient vector of logistic regression, $f(x, \theta) = \theta^T x$. We implement a first-order gradient descent optimization scheme with step size $\alpha$ to obtain the minimizer in this scenario. The $t^{th}$ iteration step is as follows:

$$\theta^{(t+1)} = \theta^{(t)} - \alpha \nabla_\theta \mathcal{L}(\theta^{(t)}) \tag{1}$$

**Parameter components** Corresponding to $x_{\shortparallel}, x_\perp$ we have $\theta_{\shortparallel}, \theta_\perp$ such that $\theta^T x = \theta_{\shortparallel}^T x_{\shortparallel} + \theta_\perp^T x_\perp$. Naturally, the notion of on/off manifold components translates to $\theta = (\theta_{\shortparallel}, \theta_\perp)$.

### 3.3.2 Two-layer linear network

For the linear network, we over-parametrize the logistic regression case with $\theta = \mathbf{A}^T w$, where $\mathbf{A} \in \mathbb{R}^{m \times D}$ is a matrix representing the weights of the first layer with $m$-neurons, and $w \in \mathbb{R}^m$ represents the weights of the output layer. $\gamma = (\text{vec } \mathbf{A}, w)$ and the model is $f(x, \gamma) = w^T \mathbf{A} x$. The parameter space is the product space $\Gamma = \mathcal{A} \times \mathcal{W}$ of the first and second layers.

As we are working with compact spaces, minimization over the product space is equivalent to sequential minimization over the first and second layer, i.e.

$$\min_{\gamma \in \mathcal{A} \times \mathcal{W}} \mathcal{L}(\gamma) = \min_{w \in \mathcal{W}} \min_{\text{vec } \mathbf{A} \in \mathcal{A}} \mathcal{L}(\gamma)$$

Consequently, we can implement an alternating gradient descent (AGD) algorithm with step sizes $\alpha_1, \alpha_2$ to obtain the minimizer. The $t^{th}$ step of AGD involves the following two gradient descent steps:

$$w\text{-step: } w^{(t+1)} = w^{(t)} - \alpha_1 \nabla_w \mathcal{L}\left(w^{(t)}, \mathbf{A}^{(t)}\right) \tag{2}$$

$$\mathbf{A}\text{-step: } \mathbf{A}^{(t+1)} = \mathbf{A}^{(t)} - \alpha_2 \nabla_\mathbf{A} \mathcal{L}\left(w^{(t+1)}, \mathbf{A}^{(t)}\right) \tag{3}$$

**Identifiability issue** For the two-layer model, the optimal parameter $\gamma^* = (\text{vec } \mathbf{A}^*, w^*)$ isn't unique, however the corresponding logistic regression coefficient $\theta^* = \mathbf{A}^{*T} w^*$ is unique and identifiable. The AGD steps in equations (2, 3) induce a sequence in $\theta$ as well, with $\theta^{(2t)} = \mathbf{A}^{(t)T} w^{(t)}$ and $\theta^{(2t+1)} = \mathbf{A}^{(t)T} w^{(t+1)}$. In the subsequent section, we can use this identification to tackle convergence rates of AGD in terms of $\theta$ and the loss $\mathcal{L}(\theta) = \mathcal{L}(w, \mathbf{A})$. Furthermore, the notion of on/off manifold parameters can be extended in the two-layer settings in terms of $\theta = \mathbf{A}^T w = (\theta_{\shortparallel}, \theta_\perp)$.

**Orthogonalization** For technical simplicity, we consider an orthogonalization step in addition to the $w$ and $\mathbf{A}$-steps. That is, before the $t^{th}$ iteration, $\mathbf{A}^{(t)}$ is column-orthogonalized such that $\mathbf{A}^{(t)T} \mathbf{A}^{(t)} = \mathbf{I}_D$, and $w^{(t)}$ is recalibrated to preserve $\theta^{(t)}$, i.e., $\mathbf{A}^{(t)T} w^{(t)}$ keeps the same after orthogonalization. However, this assumption is practical, as orthogonality improves generalizability and curbs vanishing gradient issues (Li et al., 2019; Achour et al., 2022).

## 4 Main Results

### 4.1 Motivation

Consider the expected gradient and Hessian of the loss w.r.t. the identifiable parameter $\theta$. Denote the score as $z = f(x, \gamma)$ and $\sigma(v) = (1 + \exp(-v))^{-1}$ as the standard sigmoid function.

$$\nabla_\theta \mathcal{L}(\gamma) = - \mathop{\mathbb{E}}_{\sim x,y} yx\sigma(-y \cdot z) \tag{4}$$

$$\nabla_\theta^2 \mathcal{L}(\gamma) = \mathop{\mathbb{E}}_{\sim x,y} xx^T \sigma(z)\sigma(-z) \tag{5}$$

Eq. (4) is the gradient over the distribution over $x$. Thus, for $x_{\shortparallel}$ belonging to the well-separated region, the gradients are accumulated constructively; in contrast, for all $x_{\shortparallel}$ belonging to the overlapping region, gradients from each class cancel out and accumulate destructively. This implies that as the overlapping or $\nu$ increases, we expect weaker gradients for learning $\shortparallel$ direction.

Furthermore, Eq. (5) showcases the hessian/curvature of the loss w.r.t. $\theta$. Notice that the curvature implicitly depends on $\mathbb{E}\, xx^T$, which essentially captures the covariance structure of the data. The variance in the $\shortparallel$ direction is controlled by $\sigma_{\shortparallel}^2$, inducing a larger curvature, compared to the variance in the $\perp$ direction which induces a small or flatter curvature. When implementing first-order gradient methods, the step size is bounded by the inverse of the largest curvature $\sigma_{\shortparallel}^{-2}$; As we want to change the parameters carefully, if the loss is sensitive in certain directions. However, this leads to slower learning in the flatter region, in this case, $\perp$ direction. Consequently, we expect faster convergence in the $\shortparallel$ direction and slower convergence in the $\perp$ direction, leading to a suboptimal solution with poor margins that is vulnerable to adversarial examples.

Technically, at a very high level, we bound the loss Hessian based on variance matrices derived from the data structure. Subsequently, we use Taylor expansions of the loss, Lipschitz smoothness, strong convexity and PL-inequality-based arguments to derive parameter/loss convergence rates.

We formalize the prior intuitions in the following subsection with our main theorems.

## 4.2 Theorems

For both the logistic regression and two-layer linear network case, we denote the change in loss from $\theta^t \rightarrow \theta^{t+1}$ as: $\Delta\mathcal{L}(t) = \mathcal{L}(\theta^{(t+1)}) - \mathcal{L}(\theta^{(t)})$. Furthermore, the change in loss contributed by the on/off manifold direction is denoted by $\Delta_{\shortparallel}\mathcal{L}(t) = \mathcal{L}(\theta_{\shortparallel}^{(t+1)}, \theta_{\perp}^{(t)}) - \mathcal{L}(\theta^{(t)})$ and $\Delta_{\perp}\mathcal{L}(t) = \mathcal{L}(\theta_{\shortparallel}^{(t)}, \theta_{\perp}^{(t+1)}) - \mathcal{L}(\theta^{(t)})$ respectively.

**Theorem 4.1** (Progressive bounds). *Suppose $(x, y)$ follows data distribution described in Section 3.2, then for the $t^{th}$ iterate of $\theta$ induced by GD (Eq. (1)), w-step (Eq. (2)) or **A**-step of AGD (Eq. (3)) we have:*

$$\nabla_{\theta_{\shortparallel}}\mathcal{L}(\theta^{(t)}) = -(1-\nu)\vec{c_1} \odot \mathbb{1}_d \cdot {}^{l-k}\!/_2 + \nu\vec{c_2} \odot \mathbb{1}_d \cdot {}^{k}\!/_2 \tag{6}$$

$$\nabla_{\theta_{\perp}}\mathcal{L}(\theta^{(t)}) = -\pi(\vec{c_3} \odot \mu_{\perp}^{(1)}) + (1-\pi)((\mathbb{1}_g - \vec{c_6}) \odot \mu_{\perp}^{(-1)}) \tag{7}$$
$$- (\vec{c_4} + \vec{c_5}) \odot \mathbb{1}_g \cdot {}^{\sigma_{\perp}}\!/_4$$

*Furthermore, for appropriate choice of step sizes $\alpha, \alpha_1, \alpha_2 \preceq \sigma_{\shortparallel}^{-2}$, we have:*

$$\Delta_{\shortparallel}\mathcal{L}(t) \preceq -\|\nabla_{\theta_{\shortparallel}}\mathcal{L}(\theta^{(t)})\|^2 \cdot \sigma_{\shortparallel}^{-2}; \tag{8}$$
$$\Delta_{\perp}\mathcal{L}(t) \preceq -\|\nabla_{\theta_{\perp}}\mathcal{L}(\theta^{(t)})\|^2 \cdot \sigma_{\shortparallel}^{-2}$$
$$\Delta\mathcal{L}(t) \preceq -\|\nabla_{\theta}\mathcal{L}(\theta^{(t)})\|^2 \cdot \sigma_{\shortparallel}^{-2} \tag{9}$$

*where $\vec{c_i}$ are vectors dependent on $t$ with all their elements positive and $< 1$;*

As gradient norms and the variances are positive, Eq. (8) and Eq. (9) imply that at each step of GD or AGD, the overall loss strictly decreases. In particular, the loss improves strictly in both $\perp, \shortparallel$ directions.

**Effect of $\nu$** Eq. (6) decomposes the gradient in the on-manifold direction into two components corresponding to the well-separated and overlapping regions of the on-manifold distribution, respectively. Notice that the two terms are competing with each other, and as $\nu$ (overlapping coefficient) initially increases, $\|\nabla_{\theta_{\shortparallel}}\mathcal{L}(\theta^{(t)})\|$ tends to decrease due to cancellation. Hence, the loss improvement in $\shortparallel$ direction also diminishes (Eq. (8)). With the extreme increase in $\nu$ even if $\|\nabla_{\theta_{\shortparallel}}\mathcal{L}(\theta^{(t)})\|$ is large, the classifier becomes agnostic of the original class direction, due to shift in the gradient direction favoring the overlapping component.

**Theorem 4.2** (Parameter Convergence). *Suppose $(x, y)$ follows data distribution described in Section 3.2, then for both the logistic regression and two-layer linear network, let $T$ be the number of iterations w.r.t $\theta$ induced by GD (Eq. (1)), or w-step (Eq. (2)) and **A**-step of AGD (Eq. (3)) with appropriate $\alpha, \alpha_1, \alpha_2 \preceq \sigma_{\shortparallel}^{-2}$; then:*

1. *$\|\theta_{\shortparallel}^{(T)} - \theta_{\shortparallel}^*\| \leq \delta$, when $T = \Omega(\log(\|\theta^{(0)} - \theta_{\shortparallel}^*\| \cdot \delta^{-1}))$.*

2. *$\|\theta_{\perp}^{(T)} - \theta_{\perp}^*\| \leq \delta$, when $T = \Omega((\sigma_{\shortparallel}/\sigma_{\perp})^2 \cdot \log(\|\theta^{(0)} - \theta_{\perp}^*\| \cdot \delta^{-1}))$.*

Theorem 4.2 provides the convergence rate in terms of the identifiable parameter $\theta$ in both $\perp, \shortparallel$ directions. The convergence rate in the $\shortparallel$-direction is independent of the dimensionality, whereas for $\perp$-direction the rate depends on $(\sigma_\perp/\sigma_\shortparallel)^{-1}$. Note that as $\sigma_\perp/\sigma_\shortparallel \to 0$ or $\sigma_\perp = 0$ ($x_\perp$ follows a discrete distribution), the data distribution of $x$ becomes a $d$-dimensional manifold immersed in $D$-dimension space and the time required for convergence in $\perp$-direction blows to $\infty$.

**Theorem 4.3** (Loss Convergence). *Suppose $(x, y)$ follows data distribution described in Section 3.2, then for both the logistic regression and two-layer linear network, let $T$ be the number of iterations w.r.t $\theta$ induced by GD (Eq. (1)), or w-step (Eq. (2)) and $\mathbf{A}$-step of AGD (Eq. (3)) with appropriate $\alpha, \alpha_1, \alpha_2 \preceq \sigma_\shortparallel^{-2}$. If $\theta^*$ is the optimal solution, then target error $\left(\mathcal{L}(\theta^{(T)}) - \mathcal{L}(\theta^*)\right) < \delta$ can be achieved, when:*

*1. $T = \min(r_1, r_2)$ if $\delta > C$;*

*2. $T = r_2$ if $\delta < C$,*

*where $r_1 = \Omega(\log(|\mathcal{L}(\theta^{(0)}) - \mathcal{L}(\theta^*)| \cdot (\delta - C)^{-1}))$, $r_2 = \Omega((\sigma_\shortparallel/\sigma_\perp)^2 \cdot \log(|\mathcal{L}(\theta^{(0)}) - \mathcal{L}(\theta^*)| \cdot \delta^{-1}))$ and $C = \Omega(\nu \log 2)$.*

Theorem 4.3 provides the convergence rates in terms of the loss. Additionally, it states that if the error tolerance $\delta \gg C$, we can have fast convergence rate $r_1$ independent of dimensionality ($\sigma_\perp^2/\sigma_\shortparallel^2$). However, for an arbitrarily small $\delta < C$, the convergence rate $r_2$ can be significantly slower, controlled by the dimensionality ($\sigma_\perp/\sigma_\shortparallel$) of the data. The threshold $C = \Omega(\nu \log 2)$ is essentially the minimum loss that can be attained by only training $\theta_\shortparallel$ (A.5). As $\delta \to C$, the rate $r_2$ depended on dimensionality takes over.

**Well separated on-manifold distribution**   Suppose there is no overlap, i.e., $\nu = 0$, then Theorem 4.3 tells us that we can always have a fast convergence rate independent of dimensionality for any arbitrary error-tolerance $\delta$. The on-manifold coefficients are sufficient for perfect classification, corresponding to faster convergence. In this scenario, as long as convergence in $\shortparallel$ direction is attained, the data is perfectly classifiable. Hence, the classifier can achieve robustness just based on the on-manifold direction (Fig. 2b).

**Illusion of convergence**   When $\nu$ is small, the model can attain fast convergence to a small loss value; however, to perfectly classify the data distribution, convergence in both $\perp$ and $\shortparallel$ direction is still required as $\nu > 0$ (Fig. 2ba). The model in this scenario will face adversarial vulnerability due to the poor convergence in the $\perp$ direction, even though the loss value is small.

## 5   Second-order optimization to address ill-conditioning

Our results in Section 4.2 show that slow convergence in clean training arises from the use of a step size uniformly bounded by $\sigma_\shortparallel^{-2}$ across both the $\shortparallel$ (on-manifold) and $\perp$ (off-manifold) directions. A small step size is necessary in the $\shortparallel$ direction, where curvature is large, to avoid overshooting. However, applying this same bound to the $\perp$ direction, where curvature is much smaller, leads to overly conservative updates and hence slow convergence.

To eliminate this imbalance, the step size should adapt to the local curvature: in off-manifold directions, it should scale with $\sigma_\perp^{-2}$, while in on-manifold directions it remains limited by $\sigma_\shortparallel^{-2}$. Such curvature-dependent (variable) step-size limits yield convergence rates that are independent of the variance ratio $\sigma_\perp/\sigma_\shortparallel$ (Remark A.1).

However, first-order methods cannot automatically adjust step sizes based on curvature and remain bounded by a global Lipschitz constant of the overall gradienta well-known result in the optimization literature. Even so-called adaptive first-order algorithms such as Adam or AdaGrad, which perform coordinate-wise scaling, lack true curvature awareness: their effective step size is still constrained by this global constant (Barakat & Bianchi, 2020, Theorem 2). Consequently, all first-order methods suffer from the same ill-conditioning, where off-manifold updates remain limited by $\sigma_\shortparallel^{-2}$ (Global Lipschitz constant in our framework). Now to actually employ curvature-dependent step-size limits, we consider replacing the gradient descent step in Eq. (1) with a Newton step: $\theta^{(t+1)} = \theta^{(t)} - \left(\nabla_\theta^2 \mathcal{L}(\theta^{(t)})\right)^{-1} \nabla_\theta \mathcal{L}(\theta^{(t)})$. Here, the inverse Hessian plays the role of an adaptive step size: sharp directions (large curvature) are traversed cautiously, while flat directions (small curvature) are traversed liberally. Second-order methods therefore *automatically induce curvature-dependent variable step sizes*, addressing ill-conditioning.

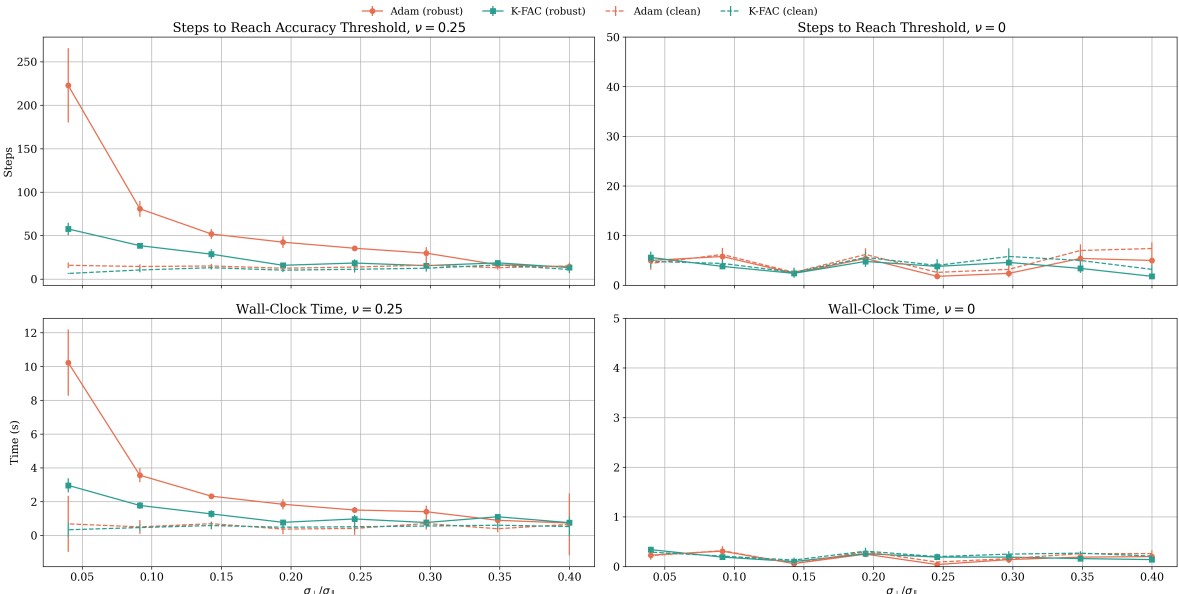

Figure 3: **Effect of dimensionality** $\sigma_\perp/\sigma_\parallel$**, on convergence.** Number of optimization steps (top) and wall-clock time (bottom) required to reach $100\%$ clean accuracy (dashed) and $95\%$ PGD-$\ell_2$ robust accuracy (solid) as the dimensionality varies. Left: data with on-manifold overlap ($\nu = 0.25$). Right: no-overlap case ($\nu = 0$). Adam's robust convergence (surrogate for optimality) becomes slow as $\sigma_\perp/\sigma_\parallel \to 0$ in the overlap regime due to ill-conditioning, whereas K-FAC mitigates this effect. In the $\nu = 0$ setting, both optimizers converge rapidly and show no dependence on dimensionality, since off-manifold convergence is unnecessary for achieving the optimal boundary.

**Practical approximations.** Computing the inverse Hessian in the Newton update is computationally prohibitive. In practice, one uses preconditioning matrices that approximate the inverse Hessian. Our experiments employ Kronecker-Factored Approximate Curvature (KFAC) (Martens & Grosse, 2015), a scalable natural-gradient method that approximates the Fisher information as a layerwise block matrix, and further factorizes each block into a Kronecker product of two smaller matrices. KFAC is computationally efficient, scales to large models, and is well-suited for distributed training.

**Convergence guarantees.** A natural question is whether second-order methods converge to the same optima as first-order methods. Recent theoretical advances suggest they do. Du et al. (2019) showed that gradient descent reliably finds global minima in sufficiently wide overparameterized networks. Zhang et al. (2019a) extended this line, proving that natural gradient descent methods (a class of second-order methods) converge to global minima as well, often with faster rates under reasonable assumptions. Moreover, they show that KFAC, as an efficient approximation, retains these convergence guarantees. These results justify our use of KFAC as a representative second-order method in our experiments.

# 6 Experiments

## 6.1 Simulation results

We empirically validate the consequences of Theorems 4.2, 4.3 and the effect of second-order KFAC optimization, using models trained on the synthetic rectangular data distribution defined in Section 3.2 (visualized in Fig. 2b). We consider two regimes: (i) a on-manifold overlap setting with $\nu = 0.25$, and (ii) a no-overlap setting with $\nu = 0$, and train using both Adam (first-order) and K-FAC (second-order) optimization.

For each configuration, we measure the number of optimization steps and wall-clock time required to reach $100\%$ clean accuracy and $95\%$ robust accuracy under $\ell_2$ Projected gradient descent (PGD)-attacks (Madry et al., 2017) of strength $\epsilon \approx |\mu_\perp^{(1)} - \mu_\perp^{(-1)}| - 2\sqrt{3}\,\sigma_\perp$, which corresponds to the effective separation between the two class clusters in the off-manifold direction. In this setting, robust accuracy serves as a proxy for proximity to the optimal decision boundary, as achieving robustness requires accurate alignment in the $\perp$-direction.

Theorem 4.2 predicts that convergence in the $\perp$-direction becomes increasingly slow as $\sigma_\perp/\sigma_\parallel \to 0$, while Theorem 4.3 implies that this slow direction affects overall loss convergence only when $\nu \neq 0$. Consequently, in the $\nu = 0.25$ regime, first-order optimization requires substantially more iterations to reach a robust solution as the dimensionality ratio decreases, since robust convergence necessitates precise off-manifold learning. In contrast, K-FAC mitigates this slowdown by incorporating curvature information. 100% Clean accuracy, however, is attained rapidly, reflecting fast convergence to a coarse loss level (Section 4.2, paragraph *Illusion of Convergence*) dependent on the extend of overlap.

In the no-overlap case $\nu = 0$, both clean and robust accuracy converge rapidly and remain insensitive to $\sigma_\perp/\sigma_\parallel$, as the optimal classifier depends solely on the on-manifold component and does not require convergence in the $\perp$-direction. Figure 3 confirms these predictions: robust convergence under Adam deteriorates sharply as $\sigma_\perp/\sigma_\parallel \to 0$ in the overlap regime, while K-FAC remains stable, whereas no such degradation appears when $\nu = 0$. With this motivation, we will move on to real-life experiments, where we show that just by long first-order training or second-order optimization we can significantly boost adversarial robustness in vision datasets without any adversarial training.

## 6.2 Computer vision datasets

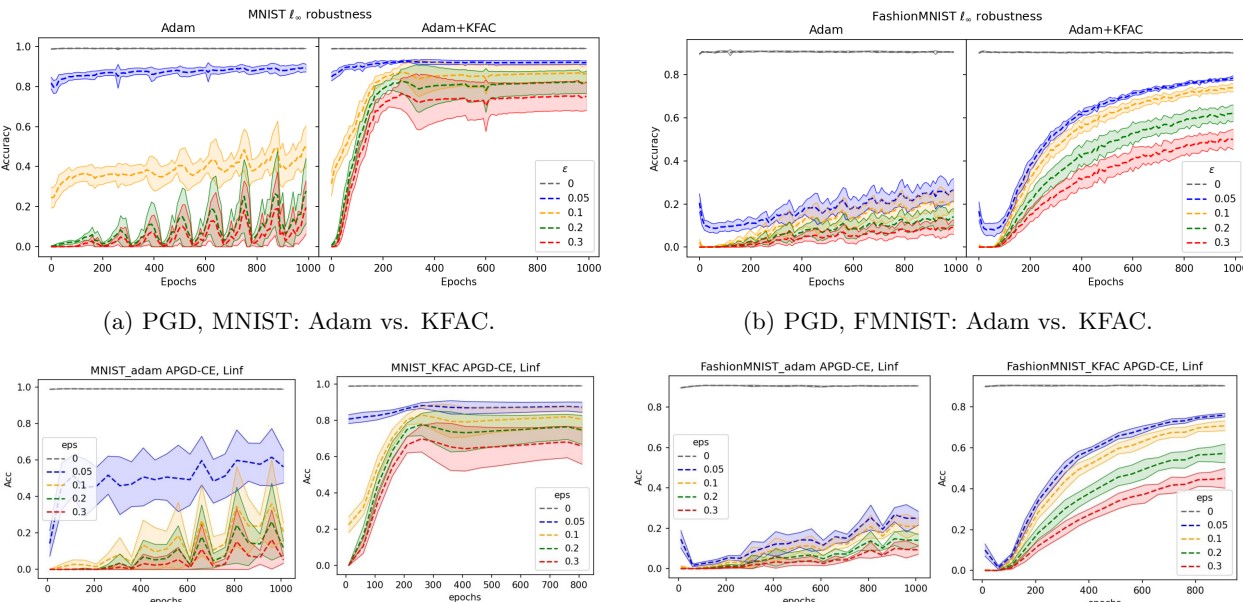

(a) PGD, MNIST: Adam vs. KFAC. (b) PGD, FMNIST: Adam vs. KFAC.

(c) APGD, MNIST, Adam. (d) APGD, MNIST, KFAC. (e) APGD, FMNIST, Adam. (f) APGD, FMNIST, KFAC.

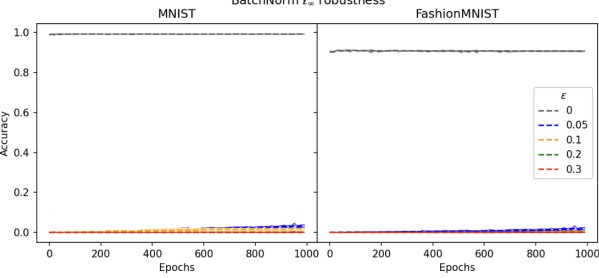

(g) PGD, BatchNorm: MNIST (left) and FMNIST (right).

Figure 4: Robustness to $\ell_\infty$ attacks over training. Top: PGD robustness, where long training increases robustness for first-order Adam and second-order KFAC reaches comparable or higher robustness much faster. Middle: APGD robustness confirms the same trend for MNIST and Fashion-MNIST, with markedly quicker gains under KFAC. Bottom: PGD robustness for BatchNorm models, where the robustness gains observed in standard networks do not appear.

Most computer vision datasets can be attributed to having a low-dimensional manifold structure (Pope et al., 2021; Osher et al., 2017). According to our framework, if there is an overlap in the manifold dimensions, the clean training is subjected to ill-conditioning, requiring considerable time to converge. Consequently, if a lack of convergence leads to adversarial vulnerability, the robust accuracy should increase with enough training. Our discussions in Section 5, imply that with a second-order optimization scheme like KFAC, this robustness improvement should be much faster, and we could attain much more robust classifiers by just using clean training. We use the cross-entropy loss in our experiments, which is a multiclass generalization of the logistic loss we used in our theoretical setup.

We perform clean training on popular computer vision datasets MNIST (LeCun et al., 2010) and FashionMNIST (Xiao et al., 2017) with a convolution neural network models (Tables 1, 3 ,LeCun et al. (2015)) under two different optimization schemes first order and second order. We use the ADAM optimizer (Kingma & Ba, 2014), which is considered one of the fastest first-order methods. We incorporate KFAC preconditioned matrices for our second-order optimization into the existing ADAM update. We use pytorch implementations (Pauloski et al., 2020; 2021) to compute the KFAC preconditioning.

At each training epoch, we subject the model to adversarial attacks to keep track of the model's robustness. We use $\ell_\infty$ Projected gradient descent (PGD)-attacks of strength $\epsilon$ to attack the models (Madry et al., 2017). We choose very small step sizes to get best approximation of PGD attacks as possible. The robust accuracy is evaluated on the test data unbeknownst to training for various choices of attack strength $\epsilon$, where $\epsilon = 0$ corresponds to the clean test accuracy. We also conduct A-PGD attacks that automatically choose the best step-sizes of PGD adaptively, and we get similar results regardless, consistent with the story line. The total training is limited to 1000 epochs for illustrative purposes. We implement 10 runs for each model to get the avg and std dev. After $\sim 10$ epochs, the clean training loss is $\sim 0$ in all scenarios. (For additional details see appendix C, Code:[1])

Figure 4 exhibits the results of our experiments described above for the MNIST and FMNIST datasets, respectively. It is evident that irrespective of the order of optimization, the robustness of the clean-trained model does increase with time under all attack strengths ($\epsilon$) with time, as suggested by our theory. Even though the clean test accuracy is stagnant around ($\epsilon = 0$) is $\sim 100, \sim 95\%$ (MNIST, FMNIST resp.) for the majority of the training, the adversarial robustness increases throughout. This validates our theory, suggesting a lack of convergence in $\perp$ direction leading to suboptimal classifiers that aren't large margin. Attaining, $\sim 0$ clean loss value and almost perfect test accuracy yet showcasing improvement in robustness throughout excessive training aligns with our discussions in section 4.2 *(Illusion of Convergence)* and the motivating illustration Fig 1 bottom right where the classifier is good enough on the data distribution, however, it hasn't attained convergence to optimal classifer and exhibits vulnerability. Additionally, the rate of robustness improvement for the second-order optimization is much faster, for the same amount of training epochs.

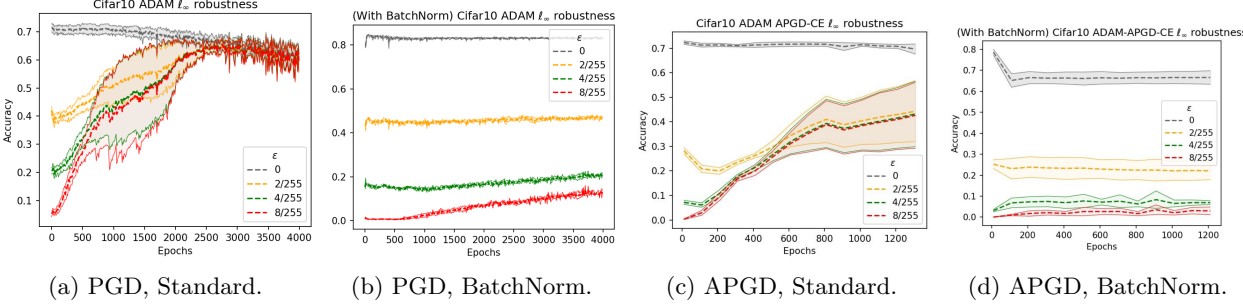

(a) PGD, Standard.  (b) PGD, BatchNorm.  (c) APGD, Standard.  (d) APGD, BatchNorm.

Figure 5: CIFAR-10 $\ell_\infty$ robustness over training for first-order optimization. PGD (left two) and APGD (right two). Standard networks (no BatchNorm) show robustness gains with longer training, while BatchNorm networks fail to exhibit similar improvements.

---

[1]https://github.com/rhaldarpurdue/Adversarial_Vulnerability_Convergence_code

Moreover, we conduct a similar robustness experiment on the CIFAR10 dataset (Krizhevsky et al., 2009) with ADAM training (Figure 5a and 5c). KFAC is designed to scale up in a distributed setting for larger models. Although we couldnt provide the KFAC version for CIFAR10 due to limited access to a single GPU, we conducted the first-order training for larger epochs to illustrate robustness improvement. We expect a second-order optimization scheme would yield similar results but with much smaller training epochs, as observed in the MNIST and FashionMNIST cases.

**Unparalleled clean training performance**   Attaining $\sim 80$ and $\sim 40\%$ robust accuracy for $\epsilon = 0.3$ (ADAM+KFAC in Fig 4) in MNIST and FMNIST datasets respectively just by clean training is unprecedented. $\epsilon = 0.3$ is a very large attack strength for these datasets, for context Madry et al. (2017) reports a robust accuracy of only  3.5% for MNIST dataset undergoing clean training with the same attack strength. Similarly, for CIFAR10 Fig 5a, we attain $\sim 60\%$ robust accuracy for $\epsilon = 8/255$ just using clean training. Traditional literature reports 0% robust accuracy for clean-trained model and 47.04% accuracy for the PGD-based adversarially trained model (Zhang et al., 2019b).

### 6.3   Vulnerability of Batch Normalization

Figures 5b, 5d, and 4g present experiments with architectures that include batch normalization (see Tables 2, 4 in the Appendix). Unlike standard networks, these models show *no robustness improvement* with extended training, even when convergence is reached. These results suggest that batch-normalized networks do not benefit from the convergence-driven robustness gains identified in Section 5.

This observation is consistent with the different implicit biases of the two architectures. Traditional ReLU networks exhibit a bias toward *maximum-margin classifiers*, which naturally promotes robustness. By contrast, batch-normalized networks have been shown to favor *uniform-margin classifiers* (Cao et al., 2023), which lack the same robustness properties. Within our framework, this helps explain why the convergence benefits we observe in standard models do not extend to batch-normalized ones.

Importantly, the vulnerability of batch normalization has also been noted in prior work, though from different perspectives. Galloway et al. (2019) reported reduced robustness, attributing it to mean-field effects. Wang et al. (2022) and Muhammad et al. (2023) showed that removing batch normalization improves robustness even under adversarial training, suggesting the effect is intrinsic to BN rather than training specifics.

Taken together, these findings indicate that the lack of robustness gains in batch-normalized models is not unique to our experiments, but part of a broader phenomenon. Our contribution is to situate this behavior within a convergence-based framework, complementing earlier explanations and raising caution about the default use of batch normalization in robustness-critical settings.

## 7   Discussion and Conclusion

We extend the notion of on and off-manifold dimensions to high and low-variance features. We explore a framework where inseparability in the on-manifold direction between data classes causes adversarial vulnerability due to slow learning in off-manifold direction from ill-conditioning. We present theoretical results supporting this hypothesis for a binary classification problem on a toy data distribution motivated by this concept. The theoretical analysis is done for the logistic regression case and 2-layer linear network for mathematical tractability. We validate the implications of our framework via simulations and real life experiments on MNIST, FMNIST, and CIFAR10 datasets with CNNs under a cross-entropy loss. Furthermore, we advocate using second-order methods that inherently circumvent ill-conditioning and accelerate convergence or continual long training for first-order method to reach closer to optimality, significantly boosting robust accuracy just by clean training.

**Adversarial Training.**   We also provide a convergence-based interpretation of adversarial training within our framework and discuss how it implicitly mitigates off-manifold ill-conditioning by inflating variance in low-variance directions. For clarity and space, we defer this discussion to Appendix B.

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

# A Proofs

In all our proofs, we assume that $Rank(\mathbf{A}) = D$, as when $Rank(\mathbf{A}) < D$, we can just work with a block of $\mathbf{A}$ that is full rank. The gradient of $w$ corresponding to the co-kernel of $\mathbf{A}$ will always be zero, and it doesn't affect any of the gradient descent steps pertaining to our analysis.

## A.1 Reparametrization

**Lemma A.1.1** (Reparametrization). *For the $t^{th}$ iterate of w-step Eq. (2) and $\mathbf{A}$-step of AGD Eq. (3) there exist reparametrizations $\tilde{w}^{(t)}$ and $\tilde{\mathbf{A}}^{(t)}$ such that $\theta^{(2t+1)} = (\theta_{\shortparallel}^{(2t+1)}, \theta_{\perp}^{(2t+1)}) = (\tilde{a}_{\shortparallel}^{(t)}, \tilde{a}_{\perp}^{(t)})$ and $\theta^{(2t)} = (\theta_{\shortparallel}^{(2t)}, \theta_{\perp}^{(2t)}) = (\tilde{w}_{\shortparallel}^{(t)}, \tilde{w}_{\perp}^{(t)})$. Where $\tilde{w} = (\tilde{w}_{\shortparallel}, \tilde{w}_{\perp}, \tilde{w}_{\vdash}), \text{vec}\, \tilde{\mathbf{A}} = (\tilde{a}_{\shortparallel}, \tilde{a}_{\perp}, \tilde{a}_{\vdash})$ and $\nabla\mathcal{L}_{\tilde{w}_{\vdash}} = \mathbf{0}_{m-D}, \nabla\mathcal{L}_{\tilde{a}_{\vdash}} = \mathbf{0}_{mD-D}$.*

*Proof.* Recall that, for the two-layer linear network, $\gamma = (\text{vec}\,\mathbf{A}, w)$ and $\theta = \mathbf{A}^T w$, with $\nabla_\theta \mathcal{L}(\gamma) = -\underset{\sim x,y}{\mathbb{E}}\, yx\sigma(-y \cdot z)$ (Eq. (4)). Our reparametrization/change of basis will help us separate the components of the first and second layer into $\shortparallel, \perp, \vdash$ directions corresponding to the identifiable parameter $\theta = (\theta_{\shortparallel}, \theta_{\perp})$. $\vdash$ components correspond to redundant directions that don't undergo any change in the AGD step.

### A.1.1 w-step

In this step of AGD, the second layer is fixed to $\mathbf{A}^{(t)}$. Given $\mathbf{A} = \mathbf{A}^{(t)}$, the gradient w.r.t. the first layer parameter $w$ is:

$$\nabla_w \mathcal{L}(\theta^{(2t)}) = -\mathbb{E}\sigma(-y \cdot z)y\mathbf{A}x \tag{10}$$

$\mathbf{A}$ is a rectangular matrix $(m \times D; m \geq D)$. $Rank(\mathbf{A}) \leq D$. Hence there exists a change of basis $\mathbf{B} \in \mathbb{R}^{m \times m}$ such that $\mathbf{A} = \mathbf{B}[\mathbf{e}_1 \dots \mathbf{e}_D]$ where $\mathbf{e}_i$ is the standard basis in $\mathbb{R}^m$ with $i^{th}$ coordinate being 1, and rest 0. This choice of standard basis for the $Im(\mathbf{A})$ loses no generality as we can always have $\tilde{\mathbf{B}} = \mathbf{BP}$ which permutes the basis in such a way so that the transformation adheres to the choice of a subset of the standard basis. In general $\mathbf{B} = \begin{bmatrix} \mathbf{A} & \mathbf{C} \end{bmatrix}$ has to take this kind of form, for any choice of $\mathbf{C} \in \mathbb{R}^{m \times m-D}$. Hence, the gradient w.r.t.can be expressed as:

$$\nabla_w \mathcal{L}(\theta^{(2t)}) = -\mathbb{E}\sigma(-y \cdot z)y\mathbf{B}\begin{pmatrix} x \\ \mathbf{0}_{m-D} \end{pmatrix} \tag{11}$$

We can choose an appropriate full rank $\mathbf{C}$ outside the span of $\mathbf{B}$ to get an inverse transformation $\mathbf{B}^{-1}$. Note in the case that $Rank(\mathbf{A}) < D$ we can just work with block inverse of the full rank part in $\mathbf{B}$; as the gradient of $w$ corresponding to the co-kernel of $\mathbf{A}$ will always be zero, so it isn't of interest.
Consider the reparametrization $\tilde{w}^{(t)} = \mathbf{B}^T w^{(t)}$, with gradients:

$$\nabla_{\tilde{w}} \mathcal{L}(\theta^{(2t)}) = \mathbf{B}^{-1} \nabla_w \mathcal{L} = -\mathbb{E}\sigma(-y \cdot z)y\mathbf{A}x = -\mathbb{E}\sigma(-y \cdot z)y\begin{pmatrix} x \\ \mathbf{0}_{m-D} \end{pmatrix} \tag{12}$$

Note that, $\theta^{(2t)} = \mathbf{A}^{(t)^T} w^{(t)} = (\mathbf{B}[\mathbf{e}_1 \dots \mathbf{e}_D])^T w^{(t)} = [\mathbf{e}_1 \dots \mathbf{e}_D]^T \tilde{w}^{(t)}$. Therefore, it follows that there exists $\tilde{w}_{\shortparallel}, \tilde{w}_{\perp}$ components corresponding to $\theta_{\shortparallel}, \theta_{\perp}$ at every $t$. Note that the remaining components $\tilde{w}_{\vdash}$ are redundant and have zero gradients.

$$\nabla_{\tilde{w}} \mathcal{L}(\theta) = -\mathbb{E}\sigma(-y \cdot z)y\begin{pmatrix} x \\ \mathbf{0}_{m-D} \end{pmatrix} \tag{13}$$

$$\nabla_{\tilde{w}_{\shortparallel}} \mathcal{L}(\theta) = -\mathbb{E}\sigma(-y \cdot z)yx_{\shortparallel} \tag{14}$$

$$\nabla_{\tilde{w}_{\perp}} \mathcal{L}(\theta) = -\mathbb{E}\sigma(-y \cdot z)yx_{\perp} \tag{15}$$

$$\nabla_{\tilde{w}_{\vdash}} \mathcal{L}(\theta) = \mathbf{0}_{m-D} \tag{16}$$

### A.1.2   A-step

In this step, the first layer is fixed to $w^{(t+1)}$. Given $w = w^{(t+1)}$, the gradient w.r.t. the second layer parameters $\mathbf{A}$ is :

$$\nabla_{\mathbf{A}}\mathcal{L} = -\mathbb{E}\sigma(-y \cdot z)ywx^T \tag{17}$$

$$\nabla_{\text{vec}\,\mathbf{A}}\mathcal{L} = -\mathbb{E}\sigma(-y \cdot z)yw \otimes x \tag{18}$$

Consider the reparametrization $\tilde{\mathbf{A}}$ such that $\mathbf{A}^{(t)} = \mathbf{U}^T\tilde{\mathbf{A}}^{(t)} = \begin{pmatrix} u_1 \\ \vdots \\ u_m \end{pmatrix}^T \tilde{\mathbf{A}}$. The columns are chosen using

Gram-Schmidt such that, $u_1 = w/\|w\|^2$ and $\langle u_j \ , \ w \rangle = 0, \|u_j\| = 1$ for all $j \neq 1$. The gradient in the re-parametrized space is then:

$$\nabla_{\tilde{\mathbf{A}}}\mathcal{L} = -\mathbb{E}\sigma(-y \cdot z)y\mathbf{U}wx^T = -\mathbb{E}\sigma(-y \cdot z)y\mathbf{e}_1 x^T \tag{19}$$

$$\nabla_{\text{vec}\,\tilde{\mathbf{A}}}\mathcal{L} = -\mathbb{E}\sigma(-y \cdot z)y\mathbf{e}_1 \otimes x \tag{20}$$

Note that,

$$\theta^{(2t+1)} = \mathbf{A}^{(t)^T}w^{(t+1)} = \left(\mathbf{U}^T\tilde{\mathbf{A}}^{(t)}\right)^T w^{t+1} = \tilde{\mathbf{A}}^{(t)^T}\mathbf{U}w^{t+1} = \tilde{\mathbf{A}}^{(t)^T}\mathbf{e}_1$$

. Therefore, it follows that there exists $a_{\shortparallel}, a_{\perp}$ components for vec $\tilde{\mathbf{A}}$ corresponding to $\theta_{\shortparallel}, \theta_{\perp}$ at every $t$. Hence, vec $\tilde{\mathbf{A}} = (a_{\shortparallel}, a_{\perp}, a_{\vdash})$ with:

$$\nabla_{a_{\shortparallel}}\mathcal{L}(\theta) = -\mathbb{E}\sigma(-y \cdot z)yx_{\shortparallel} \tag{21}$$

$$\nabla_{a_{\perp}}\mathcal{L}(\theta) = -\mathbb{E}\sigma(-y \cdot z)yx_{\perp} \tag{22}$$

$$\nabla_{a_{\vdash}}\mathcal{L}(\theta) = \mathbf{0}_{mD-D} \tag{23}$$

$$\square$$

### A.2   Hessian

**Lemma A.2.1** (Lipschitz smoothness and Strong Convexity)**.** *For any value of $\gamma$ in both logistic and two-layer-linear setting. The loss Hessians w.r.t. identifiable parameter $\theta$ can be bounded as follows:*

$$c_l \cdot \begin{pmatrix} \sigma_{\shortparallel}^2\mathbf{I}_d & \mathbf{0} \\ \mathbf{0} & \sigma_{\perp}^2\mathbf{I}_g \end{pmatrix} < \nabla_{\theta}^2\mathcal{L}(\gamma) < c_u \cdot \begin{pmatrix} \sigma_{\shortparallel}^2\mathbf{I}_d & \mathbf{0} \\ \mathbf{0} & \sigma_{\perp}^2\mathbf{I}_g \end{pmatrix} \tag{24}$$

$$c_l\sigma_{\perp}^2\mathbf{I}_g < \nabla_{\theta_{\perp}}^2\mathcal{L}(\gamma) < c_u\sigma_{\perp}^2\mathbf{I}_g \tag{25}$$

$$c_l\sigma_{\shortparallel}^2\mathbf{I}_d < \nabla_{\theta_{\shortparallel}}^2\mathcal{L}(\gamma) < c_u\sigma_{\shortparallel}^2\mathbf{I}_d \tag{26}$$

*Here $c_l, c_u > 0$ are constants. Consequently, $\mathcal{L}(\gamma)$ is convex in $\theta, \theta_{\shortparallel}, \theta_{\perp}$.*

*Proof.* The Hessian of the loss w.r.t. the identifiable parameter $\theta$ is $\nabla_{\theta}^2\mathcal{L}(\gamma) = \underset{\sim x,y}{\mathbb{E}} xx^T\sigma(z)\sigma(-z)$ [Eq. (5)]. As our analysis is restricted to the compact space, $\sigma(z)$ is strictly between 0 and 1, as $\sigma(z) = 1/0$ if and only if $\theta = \pm\infty$. Let $c_l > 0$ be the lower bound of the entropy $\sigma(z)\sigma(-z)$. Then, using positive definiteness and integrating both sides (Here $>, <$ between matrices corresponds to the difference being positive/negative definite):

$$\underset{\sim x,y}{\mathbb{E}} xx^Tc_l < \nabla_{\theta}^2\mathcal{L}(\gamma) < \underset{\sim x,y}{\mathbb{E}} xx^T$$

$$(\Sigma_x + \underset{\sim x,y}{\mathbb{E}} x \underset{\sim x,y}{\mathbb{E}} x^T) \cdot c_l < \nabla_{\theta}^2\mathcal{L}(\gamma) < \Sigma_x + \underset{\sim x,y}{\mathbb{E}} x \underset{\sim x,y}{\mathbb{E}} x^T$$

Where $\Sigma_x = \begin{pmatrix} \sigma_{\shortparallel}^2\mathbf{I}_d & \mathbf{0} \\ \mathbf{0} & \sigma_{\perp}^2\mathbf{I}_g \end{pmatrix}$ is the covariance matrix. The vector outer-product $\mathbb{E}\,x\,\mathbb{E}\,x^T$ has eigenvalues $\|\mu\|^2 = \|\mathbb{E}\,x\|^2, 0, 0 \ldots, 0$. Hence, $\mathbf{0}_D \leq \mathbb{E}\,x\,\mathbb{E}\,x^T \leq \|\mu\|^2\mathbf{I}_D$. This implies that:

$$\Sigma_x \cdot c_l < \nabla_{\theta}^2\mathcal{L}(\gamma) < \Sigma_x + \|\mu\|^2\mathbf{I}_D$$

For $0 < \|\mu\|, \sigma_{\shortparallel}, \sigma_{\perp} < \infty$. We have $\|\mu\| = c_{\shortparallel}\sigma_{\shortparallel} = c_{\perp}\sigma_{\perp}$

$$c_l \cdot \begin{pmatrix} \sigma_{\shortparallel}^2 \mathbf{I}_d & \mathbf{0} \\ \mathbf{0} & \sigma_{\perp}^2 \mathbf{I}_g \end{pmatrix} < \nabla_\theta^2 \mathcal{L}(\gamma) < \begin{pmatrix} \sigma_{\shortparallel}^2 \mathbf{I}_d & \mathbf{0} \\ \mathbf{0} & \sigma_{\perp}^2 \mathbf{I}_g \end{pmatrix} + \begin{pmatrix} c_{\shortparallel}^2 \sigma_{\shortparallel}^2 \mathbf{I}_d & \mathbf{0} \\ \mathbf{0} & c_{\perp}^2 \sigma_{\perp}^2 \mathbf{I}_g \end{pmatrix}$$

Then there exists constants $c_u = \max\{1 + c_{\shortparallel}^2, 1 + c_{\perp}^2\}$ such that

$$c_l \cdot \begin{pmatrix} \sigma_{\shortparallel}^2 \mathbf{I}_d & \mathbf{0} \\ \mathbf{0} & \sigma_{\perp}^2 \mathbf{I}_g \end{pmatrix} < \nabla_\theta^2 \mathcal{L}(\gamma) < c_u \cdot \begin{pmatrix} \sigma_{\shortparallel}^2 \mathbf{I}_d & \mathbf{0} \\ \mathbf{0} & \sigma_{\perp}^2 \mathbf{I}_g \end{pmatrix}$$

$$c_l \sigma_{\perp}^2 \mathbf{I}_g < \nabla_{\theta_{\perp}}^2 \mathcal{L}(\gamma) < c_u \sigma_{\perp}^2 \mathbf{I}_g$$

$$c_l \sigma_{\shortparallel}^2 \mathbf{I}_d < \nabla_{\theta_{\shortparallel}}^2 \mathcal{L}(\gamma) < c_u \sigma_{\shortparallel}^2 \mathbf{I}_d$$

As the above Hessians are positive definite, convexity follows. $\qquad\square$

*Proof of Theorem 4.1.* **A.3  Gradient Decomposition**

From Eq. (4) $\nabla_\theta \mathcal{L}(\gamma) = - \underset{\sim x,y}{\mathbb{E}} \, yx\sigma(-y \cdot z)$. We will tackle the $\shortparallel$ / $\perp$ components of the gradient separately.

### A.3.1  On Manifold

Consider $\nabla_{\theta_{\shortparallel}} \mathcal{L}(\gamma) = - \underset{\sim x,y}{\mathbb{E}} \, yx_{\shortparallel}\sigma(-y \cdot z)$. $\nabla_{\theta_{\shortparallel}} \mathcal{L}$ can be decomposed into $\nabla_{\theta_{\shortparallel}} \mathcal{L}^{\neg\emptyset}, \nabla_{\theta_{\shortparallel}} \mathcal{L}^{\emptyset}$, the gradients arising from overlapping and non-overlapping distribution of $x_{\shortparallel}$, with probabilities $\nu, 1 - \nu$ respectively. The distribution of $x$ corresponding to the non-overlapping part and overlapping part is $x_{\shortparallel}|_{y,\emptyset} \sim \mathcal{U}[0, (l-k) \cdot y \cdot \mathbb{1}_d]$ and $x_{\shortparallel}|_{y,\neg\emptyset} \sim \mathcal{U}[-k.y \cdot \mathbb{1}_d, 0)$ resp. One can validate this with the definition of $\nu$ (Section 3.2), $x_{\shortparallel}|_{y,\emptyset}$ contributes 0 in the computation of $\nu$, while $x_{\shortparallel}|_{y,\neg\emptyset}$ has non-zero contribution over the support.

$$\nabla_{\theta_{\shortparallel}} \mathcal{L}(\gamma) = - \underset{\sim x,y}{\mathbb{E}} \, yx_{\shortparallel}\sigma(-y \cdot z) = -(1-\nu) \cdot \underset{\sim x,y|\emptyset}{\mathbb{E}} \, yx_{\shortparallel}\sigma(-y \cdot z) - \nu \cdot \underset{\sim x,y|\neg\emptyset}{\mathbb{E}} \, yx_{\shortparallel}\sigma(-y \cdot z) \tag{27}$$

$$\underset{\sim x,y|\emptyset}{\mathbb{E}} \, yx_{\shortparallel}\sigma(-y \cdot z) = \pi \cdot \underset{\sim x|y=1,\emptyset}{\mathbb{E}} \, x_{\shortparallel}\sigma(-z) + (1-\pi) \cdot \underset{\sim x|y=-1,\emptyset}{\mathbb{E}} \, -x_{\shortparallel}\sigma(z) \tag{28}$$

$$\underset{\sim x|y=1,\emptyset}{\mathbb{E}} \, x_{\shortparallel}\sigma(-z) = \left[ \int_0^{l-k} \cdots \int_0^{l-k} x_{\shortparallel_i}\sigma(-z) \cdot (l-k)^{-d} \right] dx \quad, i \in \{1,\dots,d\} \tag{29}$$

$$= \vec{c_1} \odot \mathbb{1}_d . {}^{l-k}/2 \tag{30}$$

In Eq. (29) every integral is in a positive domain, with $\sigma(-z) \in (0,1)$ within the compact space. $0 < \int_0^{l-k} x_{\shortparallel_i}\sigma(-\theta_{\shortparallel_i}x_{\shortparallel_i}) < \int_0^{l-k} x_{\shortparallel_i} = {}^{l-k}/2$, component-wise we get Eq. (30) where $0 < \vec{c}_{1i} < 1$ for $i \in \{1,\dots,d\}$.

$$\underset{\sim x|y=-1,\emptyset}{\mathbb{E}} \, -x_{\shortparallel}\sigma(z) = \left[ \int_{-(l-k)}^0 \cdots \int_{-(l-k)}^0 -x_{\shortparallel_i}\sigma(z) \cdot (l-k)^{-d} \right] dx \quad, i \in \{1,\dots,d\}$$

$$= \left[ \int_0^{-(l-k)} \cdots \int_0^{-(l-k)} x_{\shortparallel_i}\sigma(z) \cdot (l-k)^{-d} \right] dx$$

$$= \left[ \int_0^{l-k} \cdots \int_0^{l-k} x_{\shortparallel_i}\sigma(-z) \cdot (l-k)^{-d} \right] dx \overset{Eq.~(29)}{=} \vec{c_1} \odot \mathbb{1}_d . {}^{l-k}/2 \tag{31}$$

Substituting Eq. (30) and Eq. (31) in Eq. (28) we have:

$$\underset{\sim x,y|\emptyset}{\mathbb{E}} \, yx_{\shortparallel}\sigma(-y \cdot z) = \vec{c_1} \odot \mathbb{1}_d . {}^{l-k}/2 \tag{32}$$

$$\underset{\sim x|y=1,\neg\emptyset}{\mathbb{E}} \, x_{\shortparallel}\sigma(-z) = \left[ \int_{-k}^0 \cdots \int_{-k}^0 x_{\shortparallel_i}\sigma(-z) \cdot k^{-d} \right] dx \quad, i \in \{1,\dots,d\} \tag{33}$$

Using analogous arguments as in Eq. (29)

$$= -\vec{c_2} \odot \mathbb{1}_d .^k/2 \tag{34}$$

Similarly, we have

$$\underset{\sim x|y=-1,\neg\emptyset}{\mathbb{E}} -x_{\shortparallel}\sigma(z) = -\vec{c_2} \odot \mathbb{1}_d .^k/2 \tag{35}$$

Substitute Eq. (34) and Eq. (35) in

$$\underset{\sim x,y|\neg\emptyset}{\mathbb{E}} yx_{\shortparallel}\sigma(-y \cdot z) = \pi \cdot \underset{\sim x|y=1,\neg\emptyset}{\mathbb{E}} x_{\shortparallel}\sigma(-z) + (1-\pi) \cdot \underset{\sim x|y=-1,\neg\emptyset}{\mathbb{E}} -x_{\shortparallel}\sigma(z) \tag{36}$$

to get

$$\underset{\sim x,y|\neg\emptyset}{\mathbb{E}} yx_{\shortparallel}\sigma(-y \cdot z) = -\vec{c_2} \odot \mathbb{1}_d .^k/2 \tag{37}$$

Using Eq. (32),Eq. (37) in Eq. (27) we have:

$$\nabla_{\theta_{\shortparallel}}\mathcal{L}(\gamma) = -(1-\nu) \cdot \vec{c_1} \odot \mathbb{1}_d .^{l-k}/2 + \nu \cdot \vec{c_2} \odot \mathbb{1}_d .^k/2 \tag{38}$$

### A.3.2 Off Manifold

$$\nabla_{\theta_{\perp}}\mathcal{L}(\gamma) = -\underset{\sim x,y}{\mathbb{E}} yx_{\perp}\sigma(-y \cdot z) = -\pi \cdot \underset{\sim x|y=1}{\mathbb{E}} x_{\perp}\sigma(-z) - (1-\pi) \cdot \underset{\sim x|y=-1}{\mathbb{E}} -x_{\perp}\sigma(z) \tag{39}$$

$$\underset{\sim x|y=1}{\mathbb{E}} x_{\perp}\sigma(-z) = \Big[\int_{\mu_{\shortparallel g}^{(1)}-\sqrt{3}\sigma_{\perp}}^{\mu_{\shortparallel g}^{(1)}+\sqrt{3}\sigma_{\perp}} \cdots \int_{\mu_{\shortparallel 1}^{(1)}-\sqrt{3}\sigma_{\perp}}^{\mu_{\shortparallel 1}^{(1)}+\sqrt{3}\sigma_{\perp}} x_{\perp i}\sigma(-z) \cdot (2\sqrt{3}\sigma_{\perp})^{-g}\Big] dx \quad , i \in \{1,\dots,g\} \tag{40}$$

Let $x_{\perp} = \mu_{\perp}^{(1)} + \eta$, then

$$\begin{aligned}
&= \Big[\int_{-\sqrt{3}\sigma_{\perp}}^{\sqrt{3}\sigma_{\perp}} \cdots \int_{-\sqrt{3}\sigma_{\perp}}^{\sqrt{3}\sigma_{\perp}} (\mu_{\perp}^{(1)} + \eta)_i\sigma(-z) \cdot (2\sqrt{3}\sigma_{\perp})^{-g}\Big] d\eta \\
&= \vec{c_3} \odot \mu_{\perp}^{(1)} + \Big[\int_0^{\sqrt{3}\sigma_{\perp}} \cdots \int_0^{\sqrt{3}\sigma_{\perp}} \eta_i\sigma(-z) \cdot (2\sqrt{3}\sigma_{\perp})^{-g}\Big] d\eta \\
&+ \Big[\int_{-\sqrt{3}\sigma_{\perp}}^0 \cdots \int_{-\sqrt{3}\sigma_{\perp}}^0 \eta_i\sigma(-z) \cdot (2\sqrt{3}\sigma_{\perp})^{-g}\Big] d\eta \\
&= \vec{c_3} \odot \mu_{\perp}^{(1)} + (\vec{c_4} + \vec{c_5}) \odot \mathbb{1}_g \cdot \sigma_{\perp}/4
\end{aligned} \tag{41}$$

Also,

$$\underset{\sim x|y=-1}{\mathbb{E}} -x_{\perp}\sigma(z) = \underset{\sim x|y=-1}{\mathbb{E}} -x_{\perp} + \underset{\sim x|y=-1}{\mathbb{E}} x_{\perp}\sigma(-z)$$

Following similar steps following Eq. (40) to Eq. (41)

$$= -\mu_{\perp}^{(-1)} + \vec{c_6} \odot \mu_{\perp}^{(-1)} + (\vec{c_4} + \vec{c_5}) \odot \mathbb{1}_g \cdot \sigma_{\perp}/4 \tag{42}$$

Substituting Eq. (41), Eq. (42) in Eq. (39) we have:

$$\nabla_{\theta_{\perp}}\mathcal{L}(\gamma) = -\pi\left(\vec{c_3} \odot \mu_{\perp}^{(1)}\right) - (1-\pi)\left(-(\mathbb{1}_g - \vec{c_6}) \odot \mu_{\perp}^{(-1)}\right) - (\vec{c_4} + \vec{c_5}) \odot \mathbb{1}_g \cdot \sigma_{\perp}/4$$

## A.4  Progressive Bound

### A.4.1  Loss change all directions

$\Delta\mathcal{L}(t) = \mathcal{L}(\theta^{(t+1)}) - \mathcal{L}(\theta^{(t)})$. Using Taylor's expansion of $\mathcal{L}(\theta^{(t+1)})$ around $\theta^{(t)}$ with Lipschitz smoothness Eq. (24), we have:

$$\mathcal{L}(\theta^{(t+1)}) < \mathcal{L}(\theta^{(t)}) + \nabla_\theta\mathcal{L}(\theta^{(t)})^T(\theta^{(t+1)} - \theta^{(t)}) + \frac{c_u}{2}(\theta^{(t+1)} - \theta^{(t)})^T \begin{pmatrix} \sigma_{\shortparallel}^2\mathbf{I}_d & \mathbf{0} \\ \mathbf{0} & \sigma_\perp^2\mathbf{I}_g \end{pmatrix}(\theta^{(t+1)} - \theta^{(t)})$$

As $\sigma_\perp < \sigma_{\shortparallel}$, we have:

$$\mathcal{L}(\theta^{(t+1)}) < \mathcal{L}(\theta^{(t)}) + \nabla_\theta\mathcal{L}(\theta^{(t)})^T(\theta^{(t+1)} - \theta^{(t)}) + \frac{c_u}{2}(\theta^{(t+1)} - \theta^{(t)})^T\sigma_{\shortparallel}^2\mathbf{I}_D(\theta^{(t+1)} - \theta^{(t)})$$

1. For **logistic case** the gradient step is $\theta^{(t+1)} = \theta^{(t)} - \alpha\nabla_\theta\mathcal{L}(\theta^{(t)})$

$$\mathcal{L}(\theta^{(t+1)}) < \mathcal{L}(\theta^{(t)}) - \alpha\|\nabla_\theta\mathcal{L}(\theta^{(t)})\|^2 + \alpha^2\frac{c_u\sigma_{\shortparallel}^2}{2}\|\nabla_\theta\mathcal{L}(\theta^{(t)})\|^2$$

$$\Delta\mathcal{L}(t) < -\alpha\|\nabla_\theta\mathcal{L}(\theta^{(t)})\|^2\left(1 - \alpha\cdot\frac{c_u\sigma_{\shortparallel}^2}{2}\right)$$

For step-size

$$\alpha \le \frac{1}{c_u\sigma_{\shortparallel}^2} \tag{43}$$

$$\Delta\mathcal{L}(t) < -(2c_u\sigma_{\shortparallel}^2)^{-1}\|\nabla_\theta\mathcal{L}(\theta^{(t)})\|^2$$

2. For $w-$**step** Eq. (2) we have $w^{(t/2+1)} = w^{(t/2)} - \alpha_1\nabla_w\mathcal{L}(\theta^{(t)})$. From lemma A.1.1, section:A.1.1we have reparametrization $(\theta, \mathbf{0}_{m-D}) = \tilde{w}^{(t/2)} = \mathbf{B}^Tw^{(t/2)}$. Note that $\mathbf{B} = \begin{bmatrix}\mathbf{A}^{(t/2)} & \mathbf{C}\end{bmatrix}$ for some specific choice of $\mathbf{C}$. This induces a gradient descent step in $\tilde{w}$ or $\theta$ space as follows $\tilde{w}^{(t/2+1)} = \tilde{w}^{(t/2)} - \alpha_1\mathbf{B}^T\mathbf{B}\nabla_{\tilde{w}}\mathcal{L}(\theta^{(t)})$ or $\theta^{(t+1)} = \theta^{(t)} - \alpha_1\mathbf{A}^{(t/2)^T}\mathbf{A}^{(t/2)}\nabla_\theta\mathcal{L}(\theta^{(t)})$. Hence, the Taylor expansion can be written as:

$$\mathcal{L}(\theta^{(t+1)}) < \mathcal{L}(\theta^{(t)}) - \alpha_1\nabla_\theta\mathcal{L}(\theta^{(t)})^T\mathbf{A}^{(t/2)^T}\mathbf{A}^{(t/2)}\nabla_\theta\mathcal{L}(\theta^{(t)}) + \alpha_1^2\frac{c_u\sigma_{\shortparallel}^2}{2}\|\mathbf{A}^{(t/2)^T}\mathbf{A}^{(t/2)}\nabla_\theta\mathcal{L}(\theta^{(t)})\|^2$$

let $\hat{\lambda}_\mathbf{A}, \check{\lambda}_\mathbf{A}$ be the minimum/maximum eigen values of $\mathbf{A}^{(t/2)^T}\mathbf{A}^{(t/2)}$ respectively. Then :

$$\mathcal{L}(\theta^{(t+1)}) < \mathcal{L}(\theta^{(t)}) - \alpha_1\hat{\lambda}_\mathbf{A}\|\nabla_\theta\mathcal{L}(\theta^{(t)})\|^2 + \alpha_1^2\frac{c_u\sigma_{\shortparallel}^2}{2}\check{\lambda}_\mathbf{A}^2\|\nabla_\theta\mathcal{L}(\theta^{(t)})\|^2$$

$$\Delta\mathcal{L}(t) < -\alpha_1\cdot\hat{\lambda}_\mathbf{A}\|\nabla_\theta\mathcal{L}(\theta^{(t)})\|^2\left(1 - \alpha_1\cdot\frac{\check{\lambda}_\mathbf{A}^2c_u\sigma_{\shortparallel}^2}{2\hat{\lambda}_\mathbf{A}}\right)$$

For step-size

$$\alpha_1 \le (M\cdot c_u\sigma_{\shortparallel}^2)^{-1} \tag{44}$$

Where $\check{\lambda}_\mathbf{A}^2/\hat{\lambda}_\mathbf{A} \le M$ for all $t$ and some $M < \infty$ (compact space).

$$\Delta\mathcal{L}(t) < -\hat{\lambda}_\mathbf{A}\cdot(2M\cdot c_u\sigma_{\shortparallel}^2)^{-1}\|\nabla_\theta\mathcal{L}(\theta^{(t)})\|^2$$

3. For $\mathbf{A}-\mathbf{step}$ Eq. (3) we have $\mathbf{A}^{(t+1/2)} = \mathbf{A}^{(t-1/2)} - \alpha_2 \nabla_\mathbf{A} \mathcal{L}(\theta^{(t)})$. From lemma A.1.1, section A.1.2; $\mathrm{vec}\, \mathbf{A}^{(t-1/2)} = \mathrm{vec}\, \mathbf{U}^T \tilde{\mathbf{A}}^{(t-1/2)} = (\mathbf{I}_D \otimes \mathbf{U}^T) \mathrm{vec}\, \tilde{\mathbf{A}}^{(t-1/2)}$. Let $\mathbf{K}^T = (\mathbf{I}_D \otimes \mathbf{U}^T)^{-1} = \mathbf{I}_D \otimes Diag\left(\|w^{(t+1/2)}\|^2, 1, \ldots, 1\right) \mathbf{U}$ as $\mathbf{U}^T Diag\left(\|w^{(t+1/2)}\|^2, 1, \ldots, 1\right) \mathbf{U} = \mathbf{I}_m$. This induces a gradient descent step in $\mathrm{vec}\, \tilde{\mathbf{A}}$ or $\theta$ space as follows $\mathrm{vec}\, \tilde{\mathbf{A}}^{(t+1/2)} = \mathrm{vec}\, \tilde{\mathbf{A}}^{(t-1/2)} - \alpha_2 \mathbf{K}^T \mathbf{K} \nabla_{\tilde{\mathbf{A}}} \mathcal{L}(\theta^{(t)})$ (where $\mathbf{K}^T \mathbf{K} = \mathbf{I}_D \otimes Diag\left(\|w^{(t+1/2)}\|^4, 1, \ldots, 1\right) \mathbf{I}_m$) or $\theta^{(t+1)} = \theta^{(t)} - \alpha_2 \|w^{(t+1/2)}\|^4 \nabla_\theta \mathcal{L}(\theta^{(t)})$. Following analogous steps to the derivation for the logistic case (Item 1), with step size

$$\alpha_2 = (W \cdot c_u \sigma_{\shortparallel}^2)^{-1} \tag{45}$$

, where $\|w^{(t+1/2)}\|^4 < W$ for all $t$. We get the following:

$$\Delta\mathcal{L}(t) < -\|w^{(t+1/2)}\|^4 \cdot (2 \cdot W c_u \sigma_{\shortparallel}^2)^{-1} \|\nabla_\theta \mathcal{L}(\theta^{(t)})\|^2$$

### A.4.2 Loss change on-manifold direction

$\Delta_{\shortparallel}\mathcal{L}(t) = \mathcal{L}(\theta_{\shortparallel}^{(t+1)}, \theta_\perp^{(t)}) - \mathcal{L}(\theta^{(t)})$. Using Taylor's expansion of $\mathcal{L}(\theta_{\shortparallel}^{(t+1)}, \theta_\perp^{(t)})$ around $\theta^{(t)}$ with Lipschitz smoothness Eq. (26), we have:

$$\mathcal{L}(\theta_{\shortparallel}^{(t+1)}, \theta_\perp^{(t)}) < \mathcal{L}(\theta_{\shortparallel}^{(t)}, \theta_\perp^{(t)}) + \nabla_{\theta_{\shortparallel}} \mathcal{L}(\theta^{(t)})^T (\theta_{\shortparallel}^{(t+1)} - \theta_{\shortparallel}^{(t)}) + \frac{c_u}{2}(\theta_{\shortparallel}^{(t+1)} - \theta_{\shortparallel}^{(t)})^T \sigma_{\shortparallel}^2 \mathbf{I}_D (\theta_{\shortparallel}^{(t+1)} - \theta_{\shortparallel}^{(t)})$$

1. **Logistic case:** From section A.4.1 Item 1 remember that the gradient step for logistic case is $\theta^{(t+1)} = \theta^{(t)} - \alpha \nabla_\theta \mathcal{L}(\theta^{(t)})$ with step size $\alpha \leq \frac{1}{c_u \sigma_{\shortparallel}^2}$ Eq. (43). Hence, $\theta_{\shortparallel}^{(t+1)} = \theta_{\shortparallel}^{(t)} - \alpha \nabla_{\theta_{\shortparallel}} \mathcal{L}(\theta^{(t)})$.

$$\Delta_{\shortparallel}\mathcal{L}(t) < -\alpha \|\nabla_{\theta_{\shortparallel}} \mathcal{L}(\theta^{(t)})\|^2 + \alpha^2 \frac{c_u \cdot \sigma_{\shortparallel}^2}{2} \|\nabla_{\theta_{\shortparallel}} \mathcal{L}(\theta^{(t)})\|^2$$

As $\alpha \leq \frac{1}{c_u \sigma_{\shortparallel}^2}$

$$\Delta_{\shortparallel}\mathcal{L}(t) < -(2 c_u \sigma_{\shortparallel}^2)^{-1} \|\nabla_{\theta_{\shortparallel}} \mathcal{L}(\theta^{(t)})\|^2$$

2. $w$-**step:** From section A.4.1 Item 2 remember that the gradient step induced in the identifiable parameter $\theta$, for $w$-step is $\theta^{(t+1)} = \theta^{(t)} - \alpha_1 \mathbf{A}^{(t/2)^T} \mathbf{A}^{(t/2)} \nabla_\theta \mathcal{L}(\theta^{(t)})$ with step size $\alpha_1 \leq (M \cdot c_u \sigma_{\shortparallel}^2)^{-1}$ Eq. (44). Due to orthogonalization of the first layer, $\mathbf{A}^{(t/2)^T} \mathbf{A}^{(t/2)} = \mathbf{I}_D$ and $\hat{\lambda}_\mathbf{A}, \check{\lambda}_\mathbf{A} = 1$ with $M = 1$. Hence, $\theta_{\shortparallel}^{(t+1)} = \theta_{\shortparallel}^{(t)} - \alpha_1 \nabla_{\theta_{\shortparallel}} \mathcal{L}(\theta^{(t)})$ and $\alpha_1 \leq (c_u \sigma_{\shortparallel}^2)^{-1}$. Now, mimicking the steps in the logistic case, we have:

$$\Delta_{\shortparallel}\mathcal{L}(t) < -(2 c_u \sigma_{\shortparallel}^2)^{-1} \|\nabla_{\theta_{\shortparallel}} \mathcal{L}(\theta^{(t)})\|^2$$

3. **A-Step** From section A.4.1 Item 3 remember that the gradient step induced in the identifiable parameter $\theta$, for $\mathbf{A}$-step is $\theta^{(t+1)} = \theta^{(t)} - \alpha_2 \|w^{(t+1/2)}\|^4 \nabla_\theta \mathcal{L}(\theta^{(t)})$ with step size $\alpha_2 = (W \cdot c_u \sigma_{\shortparallel}^2)^{-1}$ Eq. (45). Hence, $\theta_{\shortparallel}^{(t+1)} = \theta_{\shortparallel}^{(t)} - \alpha_2 \|w^{(t+1/2)}\|^4 \nabla_\theta \nabla_{\theta_{\shortparallel}} \mathcal{L}(\theta^{(t)})$. Now, following identical the steps as in the logistic case, $w$-step, we have:

$$\Delta_{\shortparallel}\mathcal{L}(t) < -\|w^{(t+1/2)}\|^4 \cdot (2 \cdot W c_u \sigma_{\shortparallel}^2)^{-1} \|\nabla_\theta \mathcal{L}(\theta^{(t)})\|^2$$

### A.4.3    Loss change off-manifold directions

$\Delta_\perp \mathcal{L}(t) = \mathcal{L}(\theta_{\shortparallel}^{(t)}, \theta_\perp^{(t+1)}) - \mathcal{L}(\theta^{(t)})$. Using Taylor's expansion of $\mathcal{L}(\theta_{\shortparallel}^{(t)}, \theta_\perp^{(t+1)})$ around $\theta^{(t)}$ with Lipschitz smoothness Eq. (25), we have:

$$\mathcal{L}(\theta_{\shortparallel}^{(t)}, \theta_\perp^{(t+1)}) < \mathcal{L}(\theta_{\shortparallel}^{(t)}, \theta_\perp^{(t)}) + \nabla_{\theta_\perp} \mathcal{L}(\theta^{(t)})^T (\theta_\perp^{(t+1)} - \theta_\perp^{(t)}) + \frac{c_u}{2} (\theta_\perp^{(t+1)} - \theta_\perp^{(t)})^T \sigma_\perp^2 \mathbf{I}_D (\theta_\perp^{(t+1)} - \theta_\perp^{(t)})$$

As $\sigma_\perp < \sigma_{\shortparallel}$, we have:

$$\mathcal{L}(\theta_{\shortparallel}^{(t)}, \theta_\perp^{(t+1)}) < \mathcal{L}(\theta_{\shortparallel}^{(t)}, \theta_\perp^{(t)}) + \nabla_{\theta_\perp} \mathcal{L}(\theta^{(t)})^T (\theta_\perp^{(t+1)} - \theta_\perp^{(t)}) + \frac{c_u}{2} (\theta_\perp^{(t+1)} - \theta_\perp^{(t)})^T \sigma_{\shortparallel}^2 \mathbf{I}_D (\theta_\perp^{(t+1)} - \theta_\perp^{(t)})$$

Follow identical steps corresponding to off-manifold direction as in section A.4.2 to get:

$$\text{(logistic step)} \quad \Delta_{\shortparallel} \mathcal{L}(t) < -(2 c_u \sigma_{\shortparallel}^2)^{-1} \|\nabla_{\theta_{\shortparallel}} \mathcal{L}(\theta^{(t)})\|^2$$
$$\text{(w-step)} \quad \Delta_{\shortparallel} \mathcal{L}(t) < -(2 c_u \sigma_{\shortparallel}^2)^{-1} \|\nabla_{\theta_{\shortparallel}} \mathcal{L}(\theta^{(t)})\|^2$$
$$\text{(A-step)} \quad \Delta_{\shortparallel} \mathcal{L}(t) < -\|w^{(t+1/2)}\|^4 \cdot (2 \cdot W c_u \sigma_{\shortparallel}^2)^{-1} \|\nabla_\theta \mathcal{L}(\theta^{(t)})\|^2$$

$\square$

**Lemma A.4.1.** *Let $\theta^* = (\theta_{\shortparallel}^*, \theta_\perp^*)$ be the optimal identifiable parameter minimizing $\mathcal{L}(\gamma)$ and $\theta^{(t)}$ be the $t^{th}$ iterate of $\theta$ induced by the optimization in logistic regression (Section 3.3.1) and 2-Linear Layer network setup (Section 3.3.2), then for $t_2 > t_1$ and appropriate $\alpha, \alpha_1, \alpha_2 \preceq \sigma_{\shortparallel}^{-2}$ we have:*

$$\mathcal{L}(\theta_{\shortparallel}, \theta_\perp^{(t_2)}) \leq \mathcal{L}(\theta_{\shortparallel}, \theta_\perp^{(t_1)}); \forall \theta_{\shortparallel} \tag{46}$$
$$\mathcal{L}(\theta_{\shortparallel}^{(t_2)}, \theta_\perp) \leq \mathcal{L}(\theta_{\shortparallel}^{(t_1)}, \theta_\perp); \forall \theta_\perp \tag{47}$$

*In particular, we have component-wise progressive bounds as such:*

$$\mathcal{L}(\theta_{\shortparallel}, \theta_\perp^{(t+1)}) - \mathcal{L}(\theta_{\shortparallel}, \theta_\perp^{(t)}) \leq c_p (2 c_u \sigma_\perp^2)^{-1} \|\nabla_{\theta_\perp} \mathcal{L}(\theta^{(t)})\|^2 \tag{48}$$
$$\mathcal{L}(\theta_{\shortparallel}^{(t+1)}, \theta_\perp) - \mathcal{L}(\theta_{\shortparallel}^{(t)}, \theta_\perp) \leq c_p (2 c_u \sigma_{\shortparallel}^2)^{-1} \|\nabla_{\theta_{\shortparallel}} \mathcal{L}(\theta^{(t)})\|^2 \tag{49}$$

*Proof.* We will prove for the $\perp$ direction; one can identically derive the result for the $\shortparallel$ direction. Suppose we do $T$ iteration of optimization; this induces a sequence in off-manifold parameters $\theta_\perp$. $\theta_\perp^{(0)} \to \theta_\perp^{(1)} \to \cdots \to \theta_\perp^{(T)}$. Fixing the $\shortparallel$ parameter to general $\theta_{\shortparallel}$, apply Taylor expansion with Lipschitz smoothness (lemma A.2.1) to the loss $\mathcal{L}(\theta_{\shortparallel}, \theta_\perp^{(t+1)})$ around the point $(\theta_{\shortparallel}, \theta_\perp^{(t)})$:

$$\mathcal{L}(\theta_{\shortparallel}, \theta_\perp^{(t+1)}) \leq \mathcal{L}(\theta_{\shortparallel}, \theta_\perp^{(t)}) + \nabla_{\theta_\perp} \mathcal{L}(\theta_{\shortparallel}, \theta_\perp^{(t)})^T (\theta_\perp^{(t+1)} - \theta_\perp^{(t)}) + \frac{c_u \sigma_\perp^2}{2} \|\theta_\perp^{(t+1)} - \theta_\perp^{(t)}\|^2$$

$\theta_\perp^{(t+1)} - \theta_\perp^{(t)} = -\check{\alpha} \nabla_{\theta_\perp} \mathcal{L}(\theta^{(t)})$ (Eq. (54)) for some step-size $\check{\alpha}$ depending on the situation.

$$\mathcal{L}(\theta_{\shortparallel}, \theta_\perp^{(t+1)}) \leq \mathcal{L}(\theta_{\shortparallel}, \theta_\perp^{(t)}) - \check{\alpha} \nabla_{\theta_\perp} \mathcal{L}(\theta_{\shortparallel}, \theta_\perp^{(t)})^T \nabla_{\theta_\perp} \mathcal{L}(\theta^{(t)}) + \check{\alpha}^2 \frac{c_u \sigma_\perp^2}{2} \|\nabla_{\theta_\perp} \mathcal{L}(\theta^{(t)})\|^2$$

$$\overset{(i)}{\leq} \mathcal{L}(\theta_{\shortparallel}, \theta_\perp^{(t)}) - \check{\alpha} \cdot c \|\nabla_{\theta_\perp} \mathcal{L}(\theta^{(t)})\|^2 + \check{\alpha}^2 \frac{c_u \sigma_\perp^2}{2} \|\nabla_{\theta_\perp} \mathcal{L}(\theta^{(t)})\|^2$$

$$\leq \mathcal{L}(\theta_{\shortparallel}, \theta_\perp^{(t)}) - \check{\alpha} \|\nabla_{\theta_\perp} \mathcal{L}(\theta^{(t)})\|^2 \underbrace{\left( c - \check{\alpha} \frac{c_u \sigma_\perp^2}{2} \right)}_{\overset{(ii)}{\geq} 0}$$

$$\leq \mathcal{L}(\theta_{\shortparallel}, \theta_\perp^{(t)}) - c_p (2 c_u \sigma_{\shortparallel}^2)^{-1} \|\nabla_{\theta_\perp} \mathcal{L}(\theta^{(t)})\|^2$$

$c_p = c$ when $c \geq 1$ and $c_p = 2c - 1$ when $c < 1$.

$(i)$ : $\nabla_{\theta_\perp}\mathcal{L}(\theta_{\shortparallel},\theta_\perp^{(t)}) = -\underset{\sim x,y}{\mathbb{E}}\, yx_\perp\sigma(-y\cdot z) = -\underset{\sim x,y}{\mathbb{E}}\, yx_\perp(1 + \exp{(y\theta_{\shortparallel}^T x_{\shortparallel})}\cdot\exp{(y\theta_\perp^{(t)^T} x_\perp)})^{-1}$ Let $r = \exp{(y\theta_{\shortparallel}^T x_{\shortparallel})}\cdot\exp{(-y\theta_{\shortparallel}^{(t)^T} x_{\shortparallel})}$. Then $\nabla_{\theta_\perp}\mathcal{L}(\theta_{\shortparallel},\theta_\perp^{(t)}) = -\underset{\sim x,y}{\mathbb{E}}\, yx_\perp(1 + r\exp{(y\theta_{\shortparallel}^{(t)^T} x_{\shortparallel})}\cdot\exp{(y\theta_\perp^{(t)^T} x_\perp)})^{-1}\underset{\text{As }r>0}{=}$ $-c\cdot\underset{\sim x,y}{\mathbb{E}}\, yx_\perp(1 + \exp{(y\theta_{\shortparallel}^{(t)^T} x_{\shortparallel})}\cdot\exp{(y\theta_\perp^{(t)^T} x_\perp)})^{-1} = c\cdot\nabla_{\theta_\perp}\mathcal{L}(\theta^{(t)})$ for some $c > 0$.

$(ii)$ : If $c \geq 1$ with $\breve{\alpha} \leq (c_u\sigma_{\shortparallel}^2)^{-1}$ Eq. (55), then $(ii) > 0$. If $0 < c < 1$, one can choose a strictly smaller step size still satisfying the upper bound of Eq. (55) $\breve{\alpha} \leq c\cdot(c_u\sigma_{\shortparallel}^2)^{-1} < (c_u\sigma_{\shortparallel}^2)^{-1}$ which makes $(ii) > 0$.

As $\mathcal{L}(\theta_{\shortparallel},\theta_\perp^{(t+1)}) \leq \mathcal{L}(\theta_{\shortparallel},\theta_\perp^{(t)})$, for all $t$ using induction $\mathcal{L}(\theta_{\shortparallel},\theta_\perp^{(t_2)}) \leq \mathcal{L}(\theta_{\shortparallel},\theta_\perp^{(t_1)})$ when $t_2 > t_1$. One can identically show that $\mathcal{L}(\theta_{\shortparallel}^{(t_2)},\theta_\perp) \leq \mathcal{L}(\theta_{\shortparallel}^{(t_1)},\theta_\perp)$. $\qquad\square$

## A.5 Convergence Theorems

*Proof of Theorem 4.2.* We start with the proof of $\perp$ direction; one can derive the result for $\shortparallel$ mimicking similar steps with minor alteration. $\theta^* = (\theta_{\shortparallel}^*,\theta_\perp^*)$ is the optimal identifiable parameter value which minimizes the loss and takes $T$ iteration of optimization.

**Upper bound the Gradient norm by Loss difference:** Using lemma A.4.1 with $t_2 = T, t_1 = t + 1$:

$$\mathcal{L}(\theta_{\shortparallel}^{(t)},\theta_\perp^*) \leq \mathcal{L}(\theta_{\shortparallel}^{(t)},\theta_\perp^{(t+1)})$$

Using lipschitz smoothness in $\perp$ direction we have

$$\leq \mathcal{L}(\theta_{\shortparallel}^{(t)},\theta_\perp^{(t)}) + \nabla_{\theta_\perp}\mathcal{L}(\theta^{(t)})^T(\theta_\perp^{(t+1)} - \theta_\perp^{(t)}) + \frac{c_u\sigma_\perp^2}{2}\|\theta_\perp^{(t+1)} - \theta_\perp^{(t)}\|^2$$

Gradient descent steps induce updates in $\perp$ of the form, $\theta_\perp^{(t+1)} - \theta_\perp^{(t)} = -\breve{\alpha}\nabla_{\theta_\perp}\mathcal{L}(\theta^{(t)})$

$$\leq \mathcal{L}(\theta_{\shortparallel}^{(t)},\theta_\perp^{(t)}) - \breve{\alpha}\|\nabla_{\theta_\perp}\mathcal{L}(\theta^{(t)})\|^2\left(1 - \breve{\alpha}\frac{c_u\sigma_\perp^2}{2}\right)$$

$$\implies \|\nabla_{\theta_\perp}\mathcal{L}(\theta^{(t)})\|^2 \leq \left(\mathcal{L}(\theta^{(t)}) - \mathcal{L}(\theta_{\shortparallel}^{(t)},\theta_\perp^*)\right)\left(1 - \breve{\alpha}\frac{c_u\sigma_\perp^2}{2}\right)^{-1}\cdot\breve{\alpha}^{-1} \tag{50}$$

Similarly, for the on-manifold direction

$$\implies \|\nabla_{\theta_{\shortparallel}}\mathcal{L}(\theta^{(t)})\|^2 \leq \left(\mathcal{L}(\theta^{(t)}) - \mathcal{L}(\theta_{\shortparallel}^*,\theta_\perp^{(t)})\right)\left(1 - \alpha_{\shortparallel}\frac{c_u\sigma_{\shortparallel}^2}{2}\right)^{-1}\cdot\breve{\alpha}^{-1} \tag{51}$$

Also, using strong convexity Eq. (25) we have:

$$\mathcal{L}(\theta_{\shortparallel}^{(t)},\theta_\perp^*) \geq \mathcal{L}(\theta^{(t)}) + \nabla_{\theta_\perp}\mathcal{L}(\theta^{(t)})^T(\theta_\perp^* - \theta_\perp^{(t)}) + \frac{c_l\sigma_\perp^2}{2}\|\theta_\perp^* - \theta_\perp^{(t)}\|^2$$

$$\nabla_{\theta_\perp}\mathcal{L}(\theta^{(t)})^T(\theta_\perp^{(t)} - \theta_\perp^*) \geq \mathcal{L}(\theta^{(t)}) - \mathcal{L}(\theta_{\shortparallel}^{(t)},\theta_\perp^*) + \frac{c_l\sigma_\perp^2}{2}\|\theta_\perp^* - \theta_\perp^{(t)}\|^2 \tag{52}$$

Similarly, for the on-manifold direction, strong convexity leads to:

$$\nabla_{\theta_{\shortparallel}}\mathcal{L}(\theta^{(t)})^T(\theta_{\shortparallel}^{(t)} - \theta_{\shortparallel}^*) \geq \mathcal{L}(\theta^{(t)}) - \mathcal{L}(\theta_{\shortparallel}^*,\theta_\perp^{(t)}) + \frac{c_l\sigma_{\shortparallel}^2}{2}\|\theta_{\shortparallel}^* - \theta_{\shortparallel}^{(t)}\|^2 \tag{53}$$

Also, from Eq. (43), Eq. (44), Eq. (45)

$$\breve{\alpha} = \begin{cases} \alpha & ,\alpha \leq (c_u\sigma_{\shortparallel}^2)^{-1}\text{ (logistic)} \\ \alpha_1 & ,\alpha_1 \leq (c_u\sigma_{\shortparallel}^2)^{-1}\text{ (w-step)} \\ \alpha_2\|w^{(t+1/2)}\|^4 & ,\alpha_2 \leq (W\cdot c_u\sigma_{\shortparallel}^2)^{-1}, \|w^{(t+1/2)}\|^4 < W, \forall t\text{ (A-step)} \end{cases} \tag{54}$$

$$\breve{\alpha} \le (c_u \sigma_{\shortparallel}^2)^{-1} \tag{55}$$

The parameter difference can be expressed in terms of the difference at the previous iteration:

$$\left\| \theta_\perp^{(t+1)} - \theta_\perp^* \right\|^2 = \| \theta_\perp^{(t)} - \breve{\alpha} \nabla_{\theta_\perp} \mathcal{L}(\theta^{(t)}) - \theta_\perp^* \|^2$$
$$= \| \theta_\perp^{(t)} - \theta_\perp^* \|^2 - 2\breve{\alpha} \nabla_{\theta_\perp} \mathcal{L}(\theta^{(t)})(\theta_\perp^{(t)} - \theta_\perp^*) + \breve{\alpha}^2 \| \nabla_{\theta_\perp} \mathcal{L}(\theta^{(t)}) \|^2$$

Substituting Eq. (52)

$$\le \| \theta_\perp^{(t)} - \theta_\perp^* \|^2 - 2\breve{\alpha} \left( \mathcal{L}(\theta^{(t)}) - \mathcal{L}(\theta_{\shortparallel}^{(t)}, \theta_\perp^*) + \frac{c_l \sigma_\perp^2}{2} \| \theta_\perp^{(t)} - \theta_\perp^* \|^2 \right) + \breve{\alpha}^2 \| \nabla_{\theta_\perp} \mathcal{L}(\theta^{(t)}) \|^2$$

Using Eq. (50)

$$\le \| \theta_\perp^{(t)} - \theta_\perp^* \|^2 - 2\breve{\alpha} \left( \mathcal{L}(\theta^{(t)}) - \mathcal{L}(\theta_{\shortparallel}^{(t)}, \theta_\perp^*) + \frac{c_l \sigma_\perp^2}{2} \| \theta_\perp^{(t)} - \theta_\perp^* \|^2 \right)$$
$$+ \left( \mathcal{L}(\theta^{(t)}) - \mathcal{L}(\theta_{\shortparallel}^{(t)}, \theta_\perp^*) \right) \left( 1 - \breve{\alpha} \frac{c_u \sigma_\perp^2}{2} \right)^{-1} \cdot \breve{\alpha}$$

Using Eq. (55) and $\sigma_\perp / \sigma_{\shortparallel} < 1$

$$= \| \theta_\perp^{(t)} - \theta_\perp^* \|^2 \left( 1 - \breve{\alpha} c_l \sigma_\perp^2 \right) + \underbrace{\left( \mathcal{L}(\theta_\perp^{(t)}, \theta_{\shortparallel}) - \mathcal{L}(\theta_\perp^*, \theta_{\shortparallel}) \right)}_{\le 0} \cdot \left( \underbrace{(1 - \breve{\alpha} \frac{c_u \sigma_\perp^2}{2})^{-1}}_{\le 2} - 2 \right) \cdot \breve{\alpha}$$

Using Eq. (55) and $c_l < c_u$ Eq. (24).

$$\le \| \theta_\perp^{(t)} - \theta_\perp^* \|^2 \underbrace{\left( 1 - \frac{c_l \sigma_\perp^2}{c_u \sigma_{\shortparallel}^2} \right)}_{\le 1}$$

Suppose we have $T$ total iterations then, inductively the equations accumulate over as:

$$\| \theta_\perp^{(T)} - \theta_\perp^* \|^2 \le \| \theta_\perp^{(0)} - \theta_\perp^* \|^2 \left( 1 - \frac{c_l \sigma_\perp^2}{c_u \sigma_{\shortparallel}^2} \right)^T \le \| \theta_\perp^{(0)} - \theta_\perp^* \|^2 \exp \left( -T \frac{c_l \sigma_\perp^2}{c_u \sigma_{\shortparallel}^2} \right) \tag{56}$$

Hence, we require $T \ge \frac{c_u \sigma_{\shortparallel}^2}{c_l \sigma_\perp^2} \log \left( \frac{\| \theta_\perp^{(0)} - \theta_\perp^* \|}{\delta} \right)$ for $\mathcal{O}(\delta)$ distance from $\theta_\perp^*$.

Similarly, one can derive the bound for the $\shortparallel$ direction mimicking the exact steps as $\perp$ direction but instead using analogous equations Eq. (53),Eq. (51).
$T \ge \frac{c_u \sigma_{\shortparallel}^2}{c_u \sigma_{\shortparallel}^2} \log \left( \frac{\| \theta_{\shortparallel}^{(0)} - \theta_{\shortparallel}^* \|}{\delta} \right) = \log \left( \frac{\| \theta_{\shortparallel}^{(0)} - \theta_{\shortparallel}^* \|}{\delta} \right)$ for $\mathcal{O}(\delta)$ distance from $\theta_{\shortparallel}^*$.

**Remark A.1.** *When deriving the bounds for $\| \theta_{\shortparallel}^{(0)} - \theta_{\shortparallel}^* \|$ vs $\| \theta_\perp^{(0)} - \theta_\perp^* \|$, the key point of difference is using the same step size $\breve{\alpha} \preceq \sigma_{\shortparallel}^{-2}$ even though the strong convexity constants are different, $\propto \sigma_{\shortparallel}^2$ for $\shortparallel$ direction and $\propto \sigma_\perp^2$ for $\perp$ direction. This introduces the ratio $\sigma_\perp / \sigma_{\shortparallel}$ in the $\perp$ case, while for $\shortparallel$ case the step size and strong-convexity parameter neutralize each other to 1. Hence, if we could enforce separate step size for $\perp, \shortparallel$ directions $\alpha_\perp \preceq \sigma_\perp^{-2}, \alpha_{\shortparallel} \preceq \sigma_{\shortparallel}^{-2}$. Then, we can get equivalent rates in both directions.*

$\square$

*Proof of Theorem 4.3.* We want to minimize the loss $\mathcal{L}(\theta) = \underset{\sim x, y}{\mathbb{E}} \ell(y \cdot z)$. The minimum loss that can be attained has a natural lower bound $\underset{\theta}{\min} \mathcal{L}(\theta) \ge (1 - \nu) \cdot \underset{\theta}{\min} \underset{\sim x, y | \emptyset}{\mathbb{E}} \ell(y \cdot z) + \nu \underset{\theta}{\min} \underset{\sim x, y | \neg \emptyset}{\mathbb{E}} \ell(y \cdot z)$. Suppose

we only optimize the loss w.r.t. $\theta_{\shortparallel}$ and $\theta_\perp = \mathbf{0}$, then a perfect classifier on $\shortparallel$ direction can distinguish $x_{\shortparallel}$ with probability 1 or 0 loss, but only with $1/2$ probability on $\perp$ direction. In this case, $\min_{\theta_{\shortparallel}} \mathcal{L}(\theta) \geq \nu \log 2$. In general, the classifier isn't perfect, and $\theta_\perp$ can be fixed at some default value. Hence, the lower bound is controlled by $\nu \log 2$ up to a constant $\min_{\theta_{\shortparallel}} \mathcal{L}(\theta) \geq C = \Omega(\nu \log 2)$. Consider two cases when the loss tolerance $\delta < C$ or $> C$.

**Case 1:** $\delta < C$

From the progressive bounds in proof of Theorem 4.1 for any optimization iterate (logistic, w-step or A-step) induced in the identifiable parameter $\theta$, has a decremental loss:

$$\mathcal{L}(\theta^{(t+1)}) - \mathcal{L}(\theta^{(t)}) \leq -(2c_u\sigma_{\shortparallel}^2)^{-1}\|\nabla_\theta\mathcal{L}(\theta^{(t)})\|^2 \tag{57}$$

$$-2c_l\sigma_\perp^2 \cdot \left(\mathcal{L}(\theta^{(t)}) - \mathcal{L}(\theta^*)\right) \geq -\|\nabla_\theta\mathcal{L}(\theta^{(t)})\|^2 \tag{PL-inequality}$$

*Proof of PL inequality:* The Eq. (PL-inequality) is a consequence of strong convexity. Using strong convexity in the space of identifiable parameter $\theta$ Lemma A.2.1,Eq. (24) we have:

$$\mathcal{L}(\theta) \geq \mathcal{L}(\theta^{(t)}) + \nabla_\theta\mathcal{L}(\theta^{(t)})^T(\theta - \theta^{(t)}) + \frac{c_l}{2}(\theta - \theta^{(t)})^T \begin{pmatrix} \sigma_{\shortparallel}^2\mathbf{I}_d & \mathbf{0} \\ \mathbf{0} & \sigma_\perp^2\mathbf{I}_g \end{pmatrix} (\theta - \theta^{(t)})$$

$$\mathcal{L}(\theta) \geq \mathcal{L}(\theta^{(t)}) + \nabla_\theta\mathcal{L}(\theta^{(t)})^T(\theta - \theta^{(t)}) + \frac{c_l \cdot \sigma_\perp^2}{2}\|\theta - \theta^{(t)}\|^2$$

Minimizing both sides w.r.t $\theta$, happens for $\theta = \theta^{(t)} - \nabla_\theta\mathcal{L}(\theta^{(t)})(c_l\sigma_\perp^2)^{-1}$

$$\mathcal{L}(\theta^*) \geq \mathcal{L}(\theta^{(t)}) - \|\nabla_\theta\mathcal{L}(\theta^{(t)})\|^2(2c_l\sigma_\perp^2)^{-1} \quad \square$$

Hence, from the progressive bound, we have Eq. (57) :

$$\mathcal{L}(\theta^{(t+1)}) \leq \mathcal{L}(\theta^{(t)}) - (2c_u\sigma_{\shortparallel}^2)^{-1}\|\nabla_\theta\mathcal{L}(\theta^{(t)})\|^2$$

Subtracting $\mathcal{L}(\theta^*)$ from both sides

$$\mathcal{L}(\theta^{(t+1)}) - \mathcal{L}(\theta^*) \leq \mathcal{L}(\theta^{(t)}) - \mathcal{L}(\theta^*) - (2c_u\sigma_{\shortparallel}^2)^{-1}\|\nabla_\theta\mathcal{L}(\theta^{(t)})\|^2$$

Using Eq. (PL-inequality)

$$\mathcal{L}(\theta^{(t+1)}) - \mathcal{L}(\theta^*) \leq \left(\mathcal{L}(\theta^{(t)}) - \mathcal{L}(\theta^*)\right) \cdot \left(1 - \frac{c_l\sigma_\perp^2}{c_u\sigma_{\shortparallel}^2}\right)$$

Suppose we have $T$ total iterations; then, inductively, the equations accumulate over as:

$$\mathcal{L}(\theta^{(T)}) - \mathcal{L}(\theta^*) \leq \left(\mathcal{L}(\theta^{(0)}) - \mathcal{L}(\theta^*)\right)\left(1 - \frac{c_l\sigma_\perp^2}{c_u\sigma_{\shortparallel}^2}\right)^T \leq \left(\mathcal{L}(\theta^{(0)}) - \mathcal{L}(\theta^*)\right)\exp\left(-T\frac{c_l\sigma_\perp^2}{c_u\sigma_{\shortparallel}^2}\right) \tag{58}$$

Hence, we require $T \geq \frac{c_u\sigma_{\shortparallel}^2}{c_l\sigma_\perp^2}\log\left(\left(\mathcal{L}(\theta^{(0)}) - \mathcal{L}(\theta^*)\right)\delta^{-1}\right)$ for $\mathcal{O}(\delta)$ error tolerance. This is rate $r_2$ in the thm statement.

**Case 2:** $\delta > C$

The rate $r_2$ proved in the previous case is universal and holds for this case as well. However, we can obtain a better rate in this scenario.

In this case, the loss can attain value $\delta$ solely by optimizing $\theta_{\shortparallel}$. Therefore, we will upper bound the original gradient descent loss sequence by a loss sequence solely dependant on updates of $\theta_{\shortparallel}$, and we will see that

because convergence is better on $\theta_{\shortparallel}$ direction, we can get better rates.

Note that if $\mathcal{L}(\theta) = C$, then using Jensen's inequality for convex functions: $\ell(\mathbb{E}_{\sim x,y} y \cdot z) \leq \mathcal{L}(\theta) = C$.

$$\ln\left(1 + \exp(-\mathbb{E}y\theta^T x)\right) \leq C$$
$$\exp(-\mathbb{E}y\theta^T x) \leq e^C - 1$$
$$\mathbb{E}y\theta^T x \geq \ln(\frac{1}{e^C - 1})$$
$$\mathbb{E}y(\langle \theta_{\shortparallel} , x_{\shortparallel}\rangle + \langle \theta_{\perp} , x_{\perp}\rangle) \geq \ln(\frac{1}{e^C - 1})$$
$$\pi(\langle \theta_{\shortparallel} , \mu_{\shortparallel}^{(1)}\rangle + \langle \theta_{\perp} , \mu_{\perp}^{(1)}\rangle) - (1 - \pi)(\langle \theta_{\shortparallel} , \mu_{\shortparallel}^{(-1)}\rangle + \langle \theta_{\perp} , \mu_{\perp}^{(-1)}\rangle) \geq \ln(\frac{1}{e^C - 1})$$

Note that the above can always be satisfied by $\theta_{\shortparallel} = c' \cdot \mu_{\shortparallel}^{(1)} - c'' \cdot \mu_{\shortparallel}^{(-1)}$ for appropriate choice of constants $c', c''$. Hence for a fixed $\theta_{\perp}$ there always exists some $\tilde{\theta}_{\shortparallel}$ such that $\mathcal{L}(\tilde{\theta}_{\shortparallel}, \theta_{\perp}) < C \implies \mathcal{L}(\tilde{\theta}_{\shortparallel}, \theta_{\perp}) - \mathcal{L}(\theta^*) < C$. This means fixing $\theta_{\perp} = \theta_{\perp}^{(0)}$ at initialization, there exists $\tilde{\theta}_{\shortparallel}$ as well which satisfies:

$$\min_{\theta_{\shortparallel}} \mathcal{L}(\theta_{\shortparallel}, \theta_{\perp}^{(0)}) - \mathcal{L}(\theta^*) \leq \mathcal{L}(\tilde{\theta}_{\shortparallel}, \theta_{\perp}^{(0)}) - \mathcal{L}(\theta^*) < C \tag{59}$$

Using lemma A.4.1 we have:

$$\mathcal{L}(\theta^{(T)}) \leq \mathcal{L}(\theta_{\shortparallel}^{(T)}, \theta_{\perp}^{(0)})$$

Subtracting $\mathcal{L}(\theta^*)$ both sides

$$\begin{aligned}
\mathcal{L}(\theta^{(T)}) - \mathcal{L}(\theta^*) &\leq \mathcal{L}(\theta_{\shortparallel}^{(T)}, \theta_{\perp}^{(0)}) - \mathcal{L}(\theta^*) \\
&= \mathcal{L}(\theta_{\shortparallel}^{(T)}, \theta_{\perp}^{(0)}) - \min_{\theta_{\shortparallel}} \mathcal{L}(\theta_{\shortparallel}, \theta_{\perp}^{(0)}) + \min_{\theta_{\shortparallel}} \mathcal{L}(\theta_{\shortparallel}, \theta_{\perp}^{(0)}) - \mathcal{L}(\theta^*) \\
&\leq (\delta - C) + C = \delta
\end{aligned} \tag{60}$$

Hence, if we find a $T$ for which $\mathcal{L}(\theta_{\shortparallel}^{(T)}, \theta_{\perp}^{(0)}) - \min_{\theta_{\shortparallel}} \mathcal{L}(\theta_{\shortparallel}, \theta_{\perp}^{(0)}) \leq \delta - C$ then we are done.

From lemma A.4.1,Eq. (49) we have a progressive bound on the off-manifold component as follows:

$$\mathcal{L}(\theta_{\shortparallel}^{(t+1)}, \theta_{\perp}^{(0)}) \leq \mathcal{L}(\theta_{\shortparallel}^{(t)}, \theta_{\perp}^{(0)}) - c_p(2c_u\sigma_{\shortparallel}^2)^{-1}\|\nabla_{\theta_{\shortparallel}}\mathcal{L}(\theta^{(t)})\|^2 \tag{61}$$

$$-2c_p'c_l\sigma_{\shortparallel}^2 \cdot \left(\mathcal{L}(\theta_{\shortparallel}^{(t)}, \theta_{\perp}^{(0)}) - \min_{\theta_{\shortparallel}} \mathcal{L}(\theta_{\shortparallel}, \theta_{\perp}^{(0)})\right) \geq -\|\nabla_{\theta_{\shortparallel}}\mathcal{L}(\theta^{(t)})\|^2 \tag{PL-inequality $\theta_{\shortparallel}$}$$

*Proof of PL inequality:* The Eq. (PL-inequality) is a consequence of strong convexity. Using strong convexity w.r.t. $\theta_{\shortparallel}$ Lemma A.2.1,Eq. (26) we have:

$$\mathcal{L}(\theta_{\shortparallel}, \theta_{\perp}^{(0)}) \geq \mathcal{L}(\theta_{\shortparallel}^{(t)}, \theta_{\perp}^{(0)}) + \nabla_{\theta_{\shortparallel}}\mathcal{L}(\theta_{\shortparallel}^{(t)}, \theta_{\perp}^{(0)})^T(\theta_{\shortparallel} - \theta_{\shortparallel}^{(t)}) + \frac{c_l \cdot \sigma_{\shortparallel}^2}{2}\|\theta_{\shortparallel} - \theta_{\shortparallel}^{(t)}\|^2$$

Minimizing both sides w.r.t $\theta_{\shortparallel}$, happens for $\theta_{\shortparallel} = \theta_{\shortparallel}^{(t)} - \nabla_{\theta_{\shortparallel}}\mathcal{L}(\theta_{\shortparallel}^{(t)}, \theta_{\perp}^{(0)})(c_l\sigma_{\shortparallel}^2)^{-1}$

$$\min_{\theta_{\shortparallel}} \mathcal{L}(\theta_{\shortparallel}, \theta_{\perp}^{(0)}) \geq \mathcal{L}(\theta_{\shortparallel}^{(t)}, \theta_{\perp}^{(0)}) - \|\nabla_{\theta_{\shortparallel}}\mathcal{L}(\theta_{\shortparallel}^{(t)}, \theta_{\perp}^{(0)})\|^2(2c_l\sigma_{\shortparallel}^2)^{-1}$$

From arguments like $(i)$ in proof of lemma A.4.1, we know $c_p'\|\nabla_{\theta_{\shortparallel}}\mathcal{L}(\theta_{\shortparallel}^{(t)}, \theta_{\perp}^{(0)})\|^2 = \|\nabla_{\theta_{\shortparallel}}\mathcal{L}(\theta^{(t)})\|^2$ for some proportionality constant $c_p'$

$$\min_{\theta_{\shortparallel}} \mathcal{L}(\theta_{\shortparallel}, \theta_{\perp}^{(0)}) \geq \mathcal{L}(\theta_{\shortparallel}^{(t)}, \theta_{\perp}^{(0)}) - \|\nabla_{\theta_{\shortparallel}}\mathcal{L}(\theta^{(t)})\|^2(2c_p'c_l\sigma_{\shortparallel}^2)^{-1} \quad \square$$

Hence, from the progressive bound, we have Eq. (61) :

$$\mathcal{L}(\theta_{\shortparallel}^{(t+1)}, \theta_{\perp}^{(0)}) \leq \mathcal{L}(\theta_{\shortparallel}^{(t)}, \theta_{\perp}^{(0)}) - c_p (2 c_u \sigma_{\shortparallel}^2)^{-1} \|\nabla_{\theta_{\shortparallel}} \mathcal{L}(\theta^{(t)})\|^2$$

Subtracting $\min_{\theta_{\shortparallel}} \mathcal{L}(\theta_{\shortparallel}, \theta_{\perp}^{(0)})$ from both sides

$$\mathcal{L}(\theta_{\shortparallel}^{(t+1)}, \theta_{\perp}^{(0)}) - \min_{\theta_{\shortparallel}} \mathcal{L}(\theta_{\shortparallel}, \theta_{\perp}^{(0)}) \leq \mathcal{L}(\theta_{\shortparallel}^{(t)}, \theta_{\perp}^{(0)}) - \min_{\theta_{\shortparallel}} \mathcal{L}(\theta_{\shortparallel}, \theta_{\perp}^{(0)}) - c_p (2 c_u \sigma_{\shortparallel}^2)^{-1} \|\nabla_{\theta_{\shortparallel}} \mathcal{L}(\theta^{(t)})\|^2$$

Using Eq. (PL-inequality $\theta_{\shortparallel}$)

$$\mathcal{L}(\theta_{\shortparallel}^{(t+1)}, \theta_{\perp}^{(0)}) - \min_{\theta_{\shortparallel}} \mathcal{L}(\theta_{\shortparallel}, \theta_{\perp}^{(0)}) \leq \left( \mathcal{L}(\theta_{\shortparallel}^{(t)}, \theta_{\perp}^{(0)}) - \min_{\theta_{\shortparallel}} \mathcal{L}(\theta_{\shortparallel}, \theta_{\perp}^{(0)}) \right) \cdot \left( 1 - \frac{c_p c_l c_p'}{c_u} \right)$$

Suppose we have $T$ total iterations; then, inductively, the equations accumulate over as:

$$\begin{aligned}
\mathcal{L}(\theta_{\shortparallel}^{(T)}, \theta_{\perp}^{(0)}) - \min_{\theta_{\shortparallel}} \mathcal{L}(\theta_{\shortparallel}, \theta_{\perp}^{(0)}) &\leq \left( \mathcal{L}(\theta^{(0)}) - \min_{\theta_{\shortparallel}} \mathcal{L}(\theta_{\shortparallel}, \theta_{\perp}^{(0)}) \right) \left( 1 - \frac{c_p c_l c_p'}{c_u} \right)^T \\
&\leq \left( \mathcal{L}(\theta^{(0)}) - \min_{\theta_{\shortparallel}} \mathcal{L}(\theta_{\shortparallel}, \theta_{\perp}^{(0)}) \right) \exp\left( -T \frac{c_p c_l c_p'}{c_u} \right) \\
&\leq \left( \mathcal{L}(\theta^{(0)}) - \mathcal{L}(\theta^*) \right) \exp\left( -T \frac{c_p c_l c_p'}{c_u} \right)
\end{aligned}$$

Hence, we require $T \geq \frac{c_u}{c_p c_l c_p'} \log \left( \left( \mathcal{L}(\theta^{(0)}) - \mathcal{L}(\theta^*) \right) (\delta - C)^{-1} \right)$ for $\mathcal{L}(\theta_{\shortparallel}^{(T)}, \theta_{\perp}^{(0)}) - \min_{\theta_{\shortparallel}} \mathcal{L}(\theta_{\shortparallel}, \theta_{\perp}^{(0)}) < \delta - C$.

Hence, validating the series of equations Eq. (60) and implying $\mathcal{L}(\theta^{(T)}) - \mathcal{L}(\theta^*) \leq \delta$. This gives us the rate $r_1$ in Theorem 4.3. Note that the rate $r_2$ holds regardless of $\delta$ value. The final rate is the minimum of the two rates. $\qquad\square$

## B  Convergence and Dimensionality Perspective on Adversarial Training

In practice, adversarial training (AT) is the dominant approach for inducing robustness in models (Goodfellow et al., 2014; Madry et al., 2017). Whereas clean training minimizes the loss on the original data distribution, AT minimizes the loss on a distribution of perturbed examples. These perturbations are controlled by a parameter $\epsilon$, which specifies the attack strength or radius around each training point. When $\epsilon$ is smaller than the natural *margin* between classes, AT can be viewed as clean training on an expanded distribution that covers an $\epsilon$-ball around the original data.

From the perspective of our framework, this has a key implication. Even if the original data lies on a low-dimensional manifold, the $\epsilon$-ball expansion increases the effective dimensionality: off-manifold (low-variance) directions are also perturbed, effectively inflating their variance. As a result, the variance ratio $\sigma_\perp / \sigma_\shortparallel$ increases under AT compared to clean training.

By Theorems 4.2 and 4.3, this change directly improves convergence rates in off-manifold directions, making it easier to reach solutions that depend on these features. This provides a convergence-based explanation for why adversarial training can succeed in learning robust classifiers even when clean training cannot, despite the fact that a robust classifier is also an optimal solution to the clean loss. In essence, AT alters the geometry of the training distribution in a way that mitigates ill-conditioning, narrowing the gap between theoretical optima and practical convergence.

## C  Additional Experimental Details

A learning rate of $10^{-3}$ with default ADAM parameters were used for clean training. For generating PGD attacks a step size of $\frac{2}{255}$ with $\lfloor \epsilon \cdot \frac{255}{2} \rfloor$ ($\epsilon$ is the attack strength) attack iterations and 1 restart was used. For KFAC preconditioner, the default hyper-parameters were used.

**Compute resource**: A single NVIDIA V100 gpu was used, requiring 2hrs, 3.5 hrs (MNIST, FMNIST 1000 epochs) per run for ADAM and ADAM+KFAC respectively. For CIFAR10 ADAM experiments it took  6 hrs per run (4000 epochs). Attack time included.

| Layers |
| --- |
| Conv2d(1, 16, 4, stride=2, padding=1), ReLU |
| Conv2d(16, 32, 4, stride=2, padding=1), ReLU |
| Linear(32*7*7,100), ReLU |
| Linear(100, 10) |

Table 1: NN architecture FMNIST/MNIST.

| Layers |
| --- |
| Conv2d(1, 16, 4, stride=2, padding=1), BatchNorm2d(16), ReLU |
| Conv2d(16, 32, 4, stride=2, padding=1), BatchNorm2d(32), ReLU |
| Linear(32*7*7,100), BatchNorm2d(100), ReLU |
| Linear(100, 10) |

Table 2: NN architecture FMNIST/MNIST with BN.

| Layers |
| --- |
| Conv2d(3, 128, 5, padding=2), ReLU |
| Conv2d(128, 128, 5, padding=2),ReLU, |
| MaxPool2d(2,2), Conv2d(128, 256, 3, padding=1),ReLU, |
| Conv2d(256, 256, 3, padding=1),ReLU, |
| MaxPool2d(2,2), Flatten(),Linear(256*8*8,1024),ReLU, |
| Linear(1024, 512),ReLU |
| Linear(512, 10) |

Table 3: NN architecture CIFAR10.

| Layers |
| --- |
| Conv2d(3, 128, 5, padding=2), ReLU |
| BatchNorm2d(128) |
| Conv2d(128, 128, 5, padding=2),ReLU, |
| BatchNorm2d(128), |
| MaxPool2d(2,2), Conv2d(128, 256, 3, padding=1),ReLU, |
| BatchNorm2d(256), |
| Conv2d(256, 256, 3, padding=1),ReLU, |
| BatchNorm2d(256), |
| MaxPool2d(2,2), Flatten(),Linear(256*8*8,1024),ReLU, |
| BatchNorm1d(1024) |
| Linear(1024, 512),ReLU |
| BatchNorm1d(512) |
| Linear(512, 10) |

Table 4: NN architecture CIFAR10 with BN.

