# OpenReview forum: "Adversarial Vulnerability from On-Manifold Inseparability and Poor Off-Manifold Convergence"
_TMLR — Accepted by TMLR_

### Review · Reviewer_kcM2 · 2025-10-06

**Summary Of Contributions:**

The paper explores the connection between adversarial robustness and proper model convergence. The authors define a data model that separates between in-manifold and off-manifold. The main thesis of this paper is to state that adversarial fragility corresponds to poor convergence in off-manifold directions of the data.
The paper extends the notion of on/off-manifold dimensions from redundant/useful features to high/low variance features, characterizing data distributions as low-dimensional manifolds embedded in higher-dimensional ambient spaces
For logistic regression and 2-layer linear networks on a specific uniform data distribution, the authors prove that convergence in the off-manifold is slower than the on-manifold directions causing the loss to appear to converge even if the classifier is far from optimal in the off-manifold directions.
The authors additionally propose second order optimisation (specifically KFAC) as a natural way to fix poor convergence in off-manifold directions.
They show that second order optimisation improves adversarial robustness in number of empirical settings (MNIST, FashionMNIST, and CIFAR10).

**Audience:**

Yes

**Audience Explanation:**

I find the paper interesting as it tries to relate adversarial robustness to model convergence, implicitly proposing robustness as a property of poor model convergence. I believe that this interested is shared among other members of the adversarial/robustenss ML community.

The theoretical analysis, while limited in scope, provides rigorous convergence rates relating to data geometry.

The batch normalization findings have practical implications for model design.

**Broader Impact Concerns:**

The paper does not raise significant broader impact concerns.

**Claims And Evidence:**

Yes

**Claims Explanation:**

The technical claims of the paper are supported by proofs in the Appendix. While I didn't reproduce the entirety of the proofs they seem to be technically correct and sound.
The numerical experiments are shown as the average over 10 different runs and the code used to produce the simulations is publicly available.

**Requested Changes:**

[Required] The authors should explicitly discuss the limitations of extending results from 2-layer linear networks to deep CNNs. Either they could provide theoretical analysis for at least a 2-layer ReLU network showing similar convergence behavior, or more carefully qualify the empirical results as supporting evidence for a hypothesis that requires further theoretical development.

[Required] Improve discussion of related works. The related work section is too brief given the extensive literature on adversarial robustness, with a focus on the theoretical part of it. The additional references that should be added are (but should not be limited to)

- Rice et al., "Overfitting in Adversarially Robust Deep Learning" - discusses convergence phenomena in adversarial training
- Ilyas et al., "Adversarial examples are not bugs, they are features" - discusses non-robust features, relevant for the paper's framework
- Tanay and Griffin, "A Boundary Tilting Persepective on the Phenomenon of Adversarial Examples" - proposes that adversarial examples exist when classification boundaries lie close to the data manifold and introduces the concept of "boundary tilting"; directly relevant to this paper's geometric analysis of decision boundaries and manifold structure
- Javanmard and Soltanolkotabi, "Precise Statistical Analysis of Classification  Accuracies for Adversarial Training'' -  provides insights for a mathematical model under logistic regression
- Zhang et al., Theoretically Principled Trade-off between Robustness and Accuracy - provides theoretical analysis of the accuracy-robustness tradeoff -
- Tanner et al., "A High Dimensional Statistical Model for Adversarial Training: Geometry and Trade-Offs" - the authors propose a model where utility and variance of each feature are connected (third paragraph of related works)
- Tsilivis and Kempe, "What can the neural tangent kernel tell us about adversarial robustness?" - uses NTK to study adversarial examples and shows robust features correspond to largest eigenvalues learned early in training
- Tsipras et al. "Robustness may be at odds with accuracy" - demonstrates fundamental tension between standard and robust accuracy; relevant to understanding why convergence to different features matters

[Minor] The figures should be improved and given the same sizes.

[Minor] On page 6 after "Effect of $\nu$" I believe it should be Theorem instead of Eq.

[Minor] The equation presenting the on/off-manifold gradient in Theorem 4.1 should be put as a display and not inline.

[Minor] While not always possible, it would be nice if the authors could give a simple sketch of the proof to obtain the result in Theorem 4.1 and give some intuition on what the different terms in the gradient mean

[Minor] It would have been interesting to include discussion (and maybe empirical measurement) of the difference in complexity and speed of the more advanced second order methods.

---

> ### Author Response · Authors · 2025-10-22
> **Response to Reviewer kcM2**
>
> We thank the reviewer for the positive and thoughtful assessment of our work, and for recognizing both the theoretical soundness and empirical validation. We address each point below and will incorporate the suggested revisions.
>
> ---
>
> #### **(1) Scope of Theoretical Results**
>
> We agree that our formal analysis currently covers logistic regression and two-layer linear networks, and that extension to nonlinear or deep CNNs remains open in the theoretical realm. Our goal here is to precisely isolate and prove the **mechanism**—that adversarial fragility arises from slow off-manifold convergence due to ill-conditioning—in a tractable setting, and then show a consistent manifestation of this behavior in real neural networks and classification problems.
>
> To bridge the theory–practice gap more transparently, we will revise the paper to (i) clearly state the scope and limitations, and (ii) provide a **heuristic perspective** suggesting how such behavior might generalize. In particular, one can draw attention to the **observed equivalence between neural networks and kernel SVMs** as a conceptual lens. When neural networks learn representations that effectively linearize the data into a low-dimensional, separable manifold (as illustrated in our Fig. 2b or in the data distribution assumed in our proofs), the **final classifying layers** can be *reasonably interpreted* as behaving like a kernel SVM operating in that induced feature space. Under this heuristic view, **ill-conditioning may persist after linearization**, continuing to produce differential convergence across manifold and off-manifold directions in the final layers—consistent with our theoretical mechanism.
>
> We will cite: *“On the Equivalence between Neural Network and Support Vector Machine”* (Y. Chen, W. Huang, L. M. Nguyen, T.-W. Weng). Moreover, in theoretical analyses it is standard to progressively extend results—from logistic to two-layer linear, multi-layer linear, two-layer ReLU, and ultimately homogeneous deep networks. Considering that this work is the first to approach adversarial robustness through the **manifold-convergence** lens, we begin with linear and two-layer linear settings to establish a rigorous foundation for future generalizations.
>
> ---
>
> #### **(2) Related Work**
>
> We appreciate the detailed list and agree the Related Work section should be broadened. In the revision, we will integrate and discuss:
>
> - **Feature/geometry perspectives:** Ilyas et al. (2019); Tanay & Griffin (2016)
> - **Trade-off & convergence:** Rice et al. (2020); Tsipras et al. (2018); Zhang et al. (2019); Tanner et al. (2022)
> - **Statistical viewpoints:** Javanmard & Soltanolkotabi (2022)
> - **Learning dynamics/eigen-structure:** Tsilivis & Kempe (2023)
>
> This context will clarify how our **manifold-convergence** lens complements existing explanations (non-robust features, boundary tilting, accuracy–robustness trade-offs).
>
> ---
>
> #### **(3) Theorem 4.1: Typo, Display, and Proof Sketch**
>
> - We will correct “Eq.” → “Theorem” and display the on/off-manifold gradient expression for readability.
>
> - **Proof sketch (as requested):**
>   *Theorem 4.1 follows a convex-optimization-style argument combining progressive bounds, Taylor expansions, and Lipschitz-smoothness.*
>   The aim is to **decompose the loss decrease per iterate** by bounding the loss difference via a scalar multiple of **negative gradient-norm squared**. The explicit scalars arise from the Lipschitz bound dictated by the hessian or curvature of the loss, which is related to the variance structure along on- vs. off-manifold directions.
>   This yields iterative contraction bounds of the form
>   $
>   \text{next-iterate loss} \leq \text{current loss} - c \lVert \nabla \ell \rVert^2,
>   $
>   from which the differential rates follow. Moreover, The **gradient decomposition equation** arise from **explicitly computing expectations on the data distribution. We can add a similar sketch in the final verison.
>   However, all our theorems stem from the primary motivation discussed in section 4.1.
>
> ---
>
> #### **(4) Second-Order Optimization Complexity**
>
> We can add a short discussion referring **per-iteration compute cost** (K-FAC vs. Adam) versus **epochs to target performance**. Although K-FAC incurs additional computation per step beign a second order method, it achieves **faster effective convergence**. We will include standard references contrasting these trade-offs.
>
> ---
>
> #### **(5) Minor Corrections**
>
> We will standardize figure sizes, fix typographical issues, and improve visual consistency.

---

> > ### Comment · Reviewer_kcM2 · 2025-10-22
> >
> > Thank you for the comprehensive responses to the reviews. I appreciate the effort put into addressing the concerns. Below are some follow-up points:
> >
> > **Citation Correction**
> >
> > In your response, you cite "Tanner et al. (2022)" in the related work expansion. The correct citation should be Tanner et al. (2024) based on the reference list provided.
> >
> > **Computational Complexity Analysis**
> >
> > Regarding second-order optimization complexity, while you mention adding a discussion of per-iteration compute cost, I reccomend also including a simple experiment with explicit timing comparisons for different variance ratio scenarios. Specifically one could provide wall-clock time (or equivalent metric) to reach a fixed accuracy threshold for Adam vs. K-FAC for different variance ratios.
> > This would strengthen your claim that K-FAC's benefits increase as manifold dimensionality decreases
> >
> > Except this last point, the responses are satisfactory. These additional simulations would strengthen the central claim.

---

> > > ### Author Response · Authors · 2025-10-28
> > >
> > > We sincerely thank the reviewer for the constructive follow-up and for acknowledging the adequacy of our previous responses. We address the remaining points below.
> > >
> > > ---
> > >
> > > #### **(1) Citation Correction**
> > >
> > > > *Tanner et al. (2024). “A High-Dimensional Statistical Model for Adversarial Training: Geometry and Trade-Offs.”*
> > >
> > > This correction will be reflected in the revised related work section.
> > >
> > > ---
> > >
> > > #### **(2) Computational Complexity and Timing Analysis**
> > >
> > > We believe the reviewer wants a simulation of the following kind:
> > >
> > > We plan to include a **controlled simulation** where we vary the data manifold variance ratio \( \sigma_{\perp} / \sigma_{\parallel} \) — effectively adjusting the “band thickness” of the manifold (as in Figure 2b) — and compare **Adam** versus **K-FAC** in terms of:
> > > - wall-clock time (and iteration count) to reach a fixed training accuracy threshold, and
> > > - how this timing difference scales as the manifold dimensionality decreases.
> > >
> > > This addition will empirically demonstrate that the **relative benefit of K-FAC increases with stronger ill-conditioning**, aligning with our theoretical predictions.
> > >
> > > We appreciate the reviewer’s thoughtful recommendations. Except for this additional experiment, we are glad that the reviewer finds the other responses satisfactory.

---

### Review · Reviewer_kj1y · 2025-10-07

**Summary Of Contributions:**

The authors propose a new mechanism that could contribute to adversarial vulnerability: the slow convergence of the networks in off-manifold directions (i.e., in the directions where the data variance is low). First, they discuss how adversarial vulnerability might arise from this phenomenon. Then, they propose a theoretical models where the slow convergence could occur. Finally, the authors test their intuition empirically by demonstrating a superior adversarial vulnerability in models trained with KFAC.

**Strengths**
1. The mechanisms of adversarial vulnerability are still not fully understood. And the current paper aims to bridge this gap.
2. Intuitively, the proposed explanation makes sense.
3. Theoretical results seem novel.

**Weaknesses**
1. While the authors discuss how the slow convergence could cause adversarial vulnerability, they do not prove this result theoretically. Moreover, the empirical evidence for this link is only based on Figure 2, which does not constitute a thorough evaluation.
2. While I understand the intent behind the theoretical analysis, I miss a discussion of the crucial limitations of the proposed framework. In particular, it is not clear how the absence of non-linearities affects the results.
3. I have concerns about the correctness of Theorem 4.2 (see below).
4. I do not clearly see the link between Section 4 and Sections 5 and 6 (see below).
5. Additionally, I think the work would benefit from a more thorough discussion of the literature. First, I believe it is beneficial to discuss the link with the simplicity bias of SGD (e.g., Kalimeris et al., 2019), given that the current paper shows that some complex features are learned later in training. Second, I think it is important to discuss the link with the theoretical literature on shortcut learning (e.g., Hermann et al., 2024), as the outlined mechanisms are quite similar.

**References**

Kalimeris, D., Kaplun, G., Nakkiran, P., Edelman, B., Yang, T., Barak, B., & Zhang, H. (2019). SGD on neural networks learns functions of increasing complexity.

Hermann, K., Mobahi, H., & Mozer, M. C. (2024). On the Foundations of Shortcut Learning.

**Audience:**

Yes

**Audience Explanation:**

As I outlined earlier, the underlying causes of adversarial vulnerability are still unclear. Thus, a well-supported mechanism could clarify some practically relevant questions about adversarial training of models.

**Broader Impact Concerns:**

I do not have concerns about the ethical implications of the work.

**Claims And Evidence:**

No

**Claims Explanation:**

1. About Theorem 4.2. I do not see how Equation (55) implies the lower bound on $T$: it only implies the upper bound. Given that this is one of the main results of the paper, it is crucial to clarify this implication.
2. (more minor) Additionally, given that the classification result does not depend on the scale of the parameters and only on direction, it would be more appropriate to discuss the directional convergence of $\theta$. In particular, currently it is impossible to distinguish between the two mechanisms: the domination of on-manifold parameters for classification and the incorrect direction of off-manifold parameters.
3. About Sections 5 and 6. The authors suggest that KFAC-conditioning should perform better due to the adjustment of step sizes. However, the same could be said about the Adam optimizer; in fact, the adaptivity of step-sizes to different parameters was the initial motivation behind Adam. The point of curvature conditioning is actually two-fold: choose the right learning rates for the parameters and simplify the optimization geometry. The current experiments suggest that, in fact, the geometry of the parameter space appears to play a crucial role, rather than the difference in learning speed between the parameters.

**Requested Changes:**

1. Please clarify the link between the slow convergence of off-manifold directions and adversarial vulnerability.
2. Please expand the discussion of the theoretical framework limitations.
3. Please clarify the proof of Theorem 4.2, and comment on the directional convergence.
4. Please explain the purpose and scope of Section 6. Specifically, please clarify the link between Sections 4 and 6.
5. Please add a discussion of the literature on simplicity bias, shortcut mechanisms, and expand the discussion of different mechanisms of adversarial vulnerability.

---

> ### Author Response · Authors · 2025-10-22
> **Response to Reviewer kj1y Part 1**
>
> We sincerely thank the reviewer for their thoughtful reading and for recognizing the novelty of our convergence-based perspective on adversarial vulnerability. We appreciate the constructive feedback and address each point below.
>
> #### **(1) Clarification on Theorem 4.2**
>
> **Context.**
>
> We agree that certain confusion is caused by the wording of the theorem statement. We will slightly rephrase the theorem statement in the revision as follows:
>
> The algorithm satisfies:
>
> $
> ||\\theta^{(T)}\_{\\parallel}-\\theta^{*}\_{\\parallel}||\\le\\delta
> \\quad\\text{when}\\quad T=\\Omega(\\log(||\\theta^{(0)}\_{\\parallel}-\\theta^{\\star}\_{\\parallel}||\\delta^{-1})),
> $
>
> $
> ||\\theta^{(T)}\_{\\perp}-\\theta^{*}\_{\\perp}||\\le\\delta
> \\quad\\text{when}\\quad
> T=\\Omega((\\sigma_{\\parallel}/\\sigma_{\\perp})^2\\log(||\\theta^{(0)}\_{\\parallel}-\\theta^{\\star}\_{\\parallel}||\\delta^{-1})).
> $
>
>
> **Clarification.**
> We would like to clarify that Eq. (55) represents an *intermediate contraction step* in the proof and is fully consistent with standard convergence-rate analyses in convex optimization. Specifically, we obtain
>
> $
> ||\\theta^{(T)}\_{\\perp}-\\theta^{*}\_{\\perp}||^2\\le\\|\\theta^{(0)}\_{\\perp}-\\theta^{\\star}\_{\\perp}\\|^{2}\\exp(
>  -T\\frac{c^{(\\perp)}\_{l}\\sigma\_{\\perp}^{2}}{c^{(\\parallel)}\_{u}\\sigma\_{\\parallel}^{2}})
> $
>
> which is the standard exponential-decay inequality commonly used in convex optimization analysis, i.e., linear convergence results.
> From this, one derives the **iteration complexity** given a require target accuracy level $\delta$:
>
> $
> T\\ge
> \\frac{c^{(\\parallel)}\_{u}\\sigma\_{\parallel}^{2}}{c^{(\\perp)}\_{l}\\sigma\_{\perp}^{2}}
> \\log\\!\\left(
> \\frac{\\|\\theta^{(0)}\_{\perp}-\\theta^{*}\_{\perp}\\|}{\\delta}
> \\right).
> $
>
> This formulation is equivalent to stating that the error decays as
> \\(\\|\\theta^{(T)}\_{\perp}-\\theta^{*}\_{\perp}\\|=\\mathcal{O}(\\exp(-aT))\\)
> with rate
> \\(a=\\tfrac{c^{(\\perp)}\_{l}\\sigma\_{\perp}^{2}}{c^{(\\parallel)}\_{u}\\sigma\_{\parallel}^{2}}\\),
> so achieving \\(\\mathcal{O}(\\delta)\\) accuracy accuracy is guaranteed by
> \\(T=\\Omega(\\tfrac{1}{a}\\log(1/\\delta))\\) iterations.
>
> Hence, the derivation indeed quantifies the iteration complexity to reach a target error—precisely the conventional form of convergence-rate statements in convex optimization theory.
>
> P.S., We will make an analogous clarification for **Theorem 4.3** as well.
>
> #### **(2) Misconception Regarding Adam vs. Second-Order Conditioning**
>
> We understand the reviewer’s concern that Adam also adapts step sizes. However, Adam remains a **first-order optimizer**: it relies solely on gradient information and maintains running estimates of the **first and second statistical moments of the gradients**, not curvature (i.e., the Hessian or Fisher information). Its update rule,
> $
> \theta_{t+1} = \theta_t - \eta_t \frac{m_t}{\sqrt{v_t} + \epsilon},
> $
> scales learning rates per coordinate but does **not** re-scale them according to the *true curvature* of the loss landscape.
>
> In first-order methods, the **maximum stable step size** is constrained by the inverse of the largest curvature of the loss, typically $1/L$, where $L$ is the Lipschitz constant of the gradient. In our framework, this corresponds to the inverse of the largest variance direction ($\sigma_\parallel^{-2}$).
> Although off-manifold directions have much smaller curvature ($\sigma_\perp^{-2}$), first-order optimizers must adopt a **uniform step size** bounded by the most restrictive direction for stability. This mismatch—steep on-manifold curvature vs. flat off-manifold curvature—is precisely the **ill-conditioning** that slows off-manifold convergence.
>
> This property of first-order methods is well established (e.g., *Barakat & Bianchi, 2021, Thm. 2*), showing that even adaptive methods such as Adam have effective step sizes $\Theta(1/L)$. Hence, Adam does **not** correct curvature anisotropy: despite adaptive scaling, the stable learning rate remains globally limited by the largest Hessian eigenvalue.
>
> Our analysis uses fixed-step GD up to this stability limit as the canonical first-order case—representing this behavior. The same restriction applies to momentum or adaptive variants. In experiments we include Adam (the strongest first-order baseline) to contrast it with the curvature-aware **K-FAC**.
>
> By contrast, **K-FAC** is a **second-order (curvature-aware)** optimizer. It approximates the **natural gradient** by preconditioning updates with a Kronecker-factored Fisher matrix, resolving ill-conditioning: sharp directions receive smaller steps, flatter ones larger.
>
> Although we already discuss this curvature nuance in Section 4.1, we will restate it more explicitly in Section 5 in the broader context of first-order methods to avoid confusion.

---

> ### Author Response · Authors · 2025-10-22
> **Response to Reviewer kj1y Part 2**
>
> #### **(3) Directional vs. Norm-Based Convergence**
>
> We appreciate the reviewer’s question on directional convergence. Our convergence guarantees are formulated in terms of the **parameter norm difference**, which implicitly controls both **magnitude and direction** through the triangle inequality. This form is actually stronger: it guarantees that the entire parameter vector (not just its projection) converges to the true solution. For instance, if we optimize over parameters with a fixed given magnitude the norm reduces essentially to the directional convergence result.
>
> Moreover, the classification performance here is measured in terms of the loss, and any parameter beside the optimal will have a higher loss.
>
> Intuitively, the result implies that for a fixed number of iterations $T$, the **on-manifold component** will achieve a smaller error (i.e., faster alignment with the optimal direction) than the off-manifold one. The “domination” thus reflects the faster convergence rate, not a directional misalignment.
> #### **(4) Connection Between Sections 4, 5, and 6**
>
> We will revise the narrative flow to make the logical link explicit:
> - **Section 4** establishes the *theoretical foundation*—that slow convergence in off-manifold directions is a mathematical consequence of first order GD optimization, udner data model, in optimization literature this mismatch due to curvature in first order methods is known as ill-conditioning.
> - **Section 5** Motivates the use of second order methods to resolve ill conditioning, and KFAC as a second order method, to attain convergence faster.
> - **Section 6** Empirically validate that indeed if convergence is the issue then just by using second order method or long training we should see robustness improvement by reaching closer to the optimal parameter, and we show that indeed just by long training adversarial robustness improves and for second order KFAC it is way faster. This acts as evidence to our predictions from theory.
>
>
> ---
>
> #### **(5) Discussion of Related Literature**
>
> We appreciate the reviewer’s suggestion to situate our work within the contexts of **simplicity bias** and **shortcut learning**. However, these works are addressing distinct issues even though might seem related. We will expand the related work section accordingly and clarify distinctions.
>
> - **Kalimeris et al. (2019)** show that SGD tends to learn simple (low-complexity) functions first, followed by more complex ones.
>   Our framework is distinct: we analyze *convergence rates* determined by data geometry (variance and conditioning), rather than feature complexity. In fact, even simple linear off-manifold components can exhibit slow convergence purely due to ill-conditioning (see our Fig. 2b).
>
> - **Hermann et al. (2024)** discuss shortcut learning, where models latch onto accessible but less predictive features in neural networks but the linear models are unbiased.
>   In contrast, our work focuses on the **optimization dynamics**—the model may have access to the correct features, but first-order methods converge slowly along critical low-variance directions, effectively delaying the discovery of robust solutions even when the target classifier is simple and linearly separable. Our theory works for linear models as well, logistic regression/
>
> We will incorporate both citations and highlight how these works are distinct from our convergence-based explanation.

---

> ### Comment · Reviewer_kj1y · 2025-10-26
>
> Thank you for your clarification!
>
> While I appreciate the response, I still have some concerns regarding the submission.
>
> 1. While the new formulation of the theorem is correct, it becomes a sufficient condition, not a necessary one. Thus, it is unclear whether this result reveals a fundamental hardness of optimization or is simply an artifact of the chosen proof technique.
> 2. While I understand that there are bounds on the learning rate parameter, the speed of learning across the directions would still be different. Thus, it is still unclear to me why Adam could not correct for the mechanism under analysis. (Moreover, I can not precisely understand which work the authors cited.)
> 3. I understand that norm convergence is stronger, but this fact actually weakens the result. The paper provides "lower bounds" for the stronger version of convergence, which is not directly related to the classifier performance. Again, it goes in the direction of a sufficient condition, not a necessary one.

---

> ### Author Response · Authors · 2025-10-28
>
> We thank the reviewer for inciting a nuanced discussion and would like to clarify the raised points below:
> #### **(1) On “Sufficiency” vs. “Necessity” of the Bound**
>
> We agree that the convergence bounds we derive are *sufficient* conditions — this is indeed standard in optimization theory (when comparing the speed of convergence).
> Our proof follows classical convergence-rate analyses used in convex and smooth non-convex optimization, where the goal is to establish iteration complexity guarantees (sufficient conditions) to reach a target error level.
>
> Importantly, our off- vs. on-manifold discrepancy does **not** arise from proof artifacts or special tricks — it emerges directly from the **difference in curvature** and the **step-size constraint** intrinsic to first-order methods.
> In other words, the slower rate along off-manifold directions is a structural outcome of ill-conditioning, not a peculiarity of our derivation.
> The rates characterize *fundamental limitations* of first-order optimization under anisotropic curvature.
>
> On the other hand, it is worth mentioning that a comparison with the necessary bound doesn't necessarily imply a stronger assertion about the convergence speed difference, as it will also face the same issue of technicality (i.e., a small necessary bound can be an artifact of poor proof techniques).
>
> ---
>
> #### **(2) On Why Adam Cannot Fully Correct the Mechanism**
>
> We appreciate the reviewer’s persistence on this point and agree it deserves deeper clarification.
> The key insight is geometric:
> if optimization occurred **only** along the off-manifold direction, the Lipschitz constant $L_{\perp}\propto\sigma_{\perp}^2$ would be very small, so the critical step size $1/L_{\perp}$ could be large.
> However, in the joint optimization over both directions, the global step size is limited by the **largest curvature** (on-manifold), i.e., $L\propto\sigma_{\parallel}^2$.
> Thus, even though the off-manifold directions would benefit from larger steps, first-order optimizers must adopt a **global step-size cap** $\alpha \le c/L_{\parallel}$, which is overly conservative in flatter directions.
>
> Adam adapts coordinate-wise scaling but remains a **first-order method** that lacks curvature awareness.
> Its effective step size is still bounded by $O(1/L)$, where $L$ is the global Lipschitz constant of the gradient.
> This property has been rigorously shown in *Barakat & Bianchi (2021, Theorem 2)*, which proves that adaptive algorithms such as Adam have a maximal stable step size bounded by $C/L$.
> Hence, Adam cannot correct for curvature anisotropy—it rescales gradients based on magnitude, not geometry.
>
> In contrast, **second-order or curvature-aware methods (e.g., K-FAC)** use explicit curvature information to *recondition the parameter space*.
> They can adapt step sizes per eigen-direction of the Hessian or Fisher matrix, enabling large steps in flat directions and small ones in sharp directions, thereby removing ill-conditioning.
> This geometric correction is fundamentally unavailable to first-order schemes.
>
> **Reference:**
> [1] *Anas Barakat & Pascal Bianchi (2021). Convergence Analysis of a Momentum Algorithm with Adaptive Step Size for Non-Convex Optimization.* [arXiv:1911.07596](https://arxiv.org/abs/1911.07596v2)
>
> ---
>
> #### **(3) On Norm-Based vs. Directional Convergence**
>
> Norm convergence is a *stronger statement*—it controls both magnitude and direction simultaneously, guaranteeing convergence of the entire parameter vector.
> Directional convergence can be derived as a special case by restricting the optimization to the unit-norm parameter space ($\Gamma:||\theta||=1$) in section 3.3.
>
> Classification performance (i.e., high classification accuracy with robustness) is a consequence of the model’s parameters approaching the **true optimum** (i.e., the minimizer of the loss), not just on alignment along a subspace.
> In Figure 2b, this is illustrated by the “illusion of convergence”: the model achieves near-perfect clean accuracy early in training, yet the off-manifold parameters have not converged, leaving the decision boundary fragile to adversarial perturbations and low adversarial accuracy.
> Only when the full parameter (both on- and off-manifold components) converges will the classifier reach the robust optimum.
>
> Hence, while the theorem provides a sufficient condition, it remains directly relevant to robustness:
> the slower off-manifold convergence explains why models can appear well-trained (low loss, high accuracy) yet remain adversarially brittle.

---

### Review · Reviewer_j4Br · 2025-10-12

**Summary Of Contributions:**

### Summary

This paper makes contributions to the theoretical foundation of adversarial examples. Specifically, the paper attributes the existence of adversarial examples to the poor convergence in off-manifold directions. With a specific data distribution and model architecture, the paper presents several theorems on the convergence of optimization along directions parallel (i.e., on-manifold) or perpendicular (i.e., off-manifold) to the data manifold. In particular, the convergence along the off-manifold direction was slower by a factor defined by the ratio of the on-manifold variance to the off-manifold variance. Consequently, the paper concludes that variable step sizes (across directions) are required to balance convergence speed and claims that the second-order optimization method already adjusts them. Through experiments, the paper demonstrates the robustness of a KFAC technique (which approximates Hessian computation) compared to naive first-order optimization. The experiments also show that the proposed idea does not improve robustness when the commonly used batch normalization is used.

### Strengths
1. Still, adversarial machine learning lacks theoretical explanations about why adversarial samples exist. This paper makes a valuable contribution towards them.
2. The paper presents novel insights into the relationship between on-manifold overlapping and adversarial robustness, showing that there may be an intrinsic vulnerability factor that depends on the data distribution.
3. The experimental results support the paper’s claim well.

### Weaknesses
I don’t find any major flaws or weaknesses in this paper. There are some questions or minor changes listed in the **Requested Changes** section.

**Additional Comments:**

This paper contains an interesting theoretical explanation of the adversarial example vulnerability. Compared to other theory papers, this paper is of good quality and not too difficult to read. While more research would be needed in this direction, e.g., how to measure the overlap in the training data, this paper provides valuable insights into a new understanding of adversarial examples.

**Audience:**

Yes

**Audience Explanation:**

To the best of my knowledge, this is the first paper that theoretically investigates the overlap along the on-manifold direction as a potential reason for the adversarial vulnerability. The theoretical contribution indicates that vulnerability may be intrinsic to the data manifold geometry, and that special care may be required during training to alleviate ill-conditioning. Practitioners would be interested in this finding to improve the robustness of their training in general.

**Broader Impact Concerns:**

I don’t see a particular broader impact concern regarding this paper.

**Claims And Evidence:**

Yes

**Claims Explanation:**

In the Appendix, the paper provides the proofs for all the theoretical results. The experiments clearly demonstrate the improved robustness achieved by employing KFAC during training.

**Requested Changes:**

Please go through the paper once again and make sure that the formulae have no typos.
* **Section 3.2**. Overlapping coefficient. There is a missing parenthesis in the equation for $\nu$.
* **Section 4.2**. Effect of $\nu$. Eq. (4.1) should be Eq. (6) in Theorem 4.2

---

> ### Author Response · Authors · 2025-10-22
> **Response to Reviewer j4Br**
>
> We sincerely thank the reviewer for the positive evaluation and for recognizing the paper’s theoretical contribution and practical relevance. We appreciate the specific suggestions and will incorporate the requested clarifications and fixes.
>
> ---
>
> #### **Requested Changes & Our Actions**
>
> 1. **Typos / Formula Consistency**
>    - We will perform a full proofreading pass to ensure formulae are typo-free and consistently formatted across the main text and appendix.
>
> 2. **Section 3.2 — Overlapping Coefficient**
>    - We will correct the **missing parenthesis** in the overlapping-coefficient formula and verify all parentheses/brackets in that subsection. We will also add a brief clarifying sentence on notation to avoid ambiguity.
>
> 3. **Section 4.2 — Cross-Reference Fix**
>    - We will fix the reference “Eq. (4.1)” to **Theorem 4.1 (Eq. (6))**.
>
> ---
>
> #### **Additional Clarifications**
>
> - **Future Directions.** We agree it would be valuable to explore **practical estimators of overlap** in real datasets. That aspect in itself could be an interesting work. Quantifying on-manifold overlap in image data is  non-trivial and requires further research. However, our goal with this work was to provide a theoretical framework and empirically show that convergence alone can affect robustness significantly.

---

### Decision · Action_Editor_HLi8 · 2025-11-17

**Recommendation:** Accept with minor revision

**Additional Comments:**

The post-review exchanges with the reviewers led to several suggestions for improvements (presentation, clarification of results, additional bibliographic references...) -that should be taken into account by the authors- that will greatly improve the clarity and comprehensiveness of the paper. The authors have acknowledged these changes -I am taking this occasion to thank all reviewers for their valuable work- and I request the author to prepare a detailed revision statement when submitting the revised version of the paper.

**Audience:**

Yes

**Audience Explanation:**

Again, very convergent assessments from the reviewers on that point. The contribution is timely and deals with an important topic; the ideas put forward in the paper are likely to inspire future research on the topic.

**Claims And Evidence:**

Yes

**Claims Explanation:**

The reviewers' recommendations are remarkably convergent regarding this paper. The main contribution of the paper is to provide models and results that support the claim that exposition to adversarial vulnerability can result from poor convergence in off-manifold directions of the data (and can, accordingly be mitigated by scaling mechanisms that tend to reduce this unbalanced convergence). All reviewers consider that the elements put forward in the paper make a convincing step in that direction and provide relevant ground for subsequent research on the topic.

---

> ### Author Response · Authors · 2025-11-23
>
> We sincerely thank the Action Editor and all reviewers for their thoughtful feedback, constructive suggestions, and engagement throughout the review process.
> We are very pleased with the positive decision and will incorporate all recommended clarifications, bibliographic additions, and presentation improvements in the final camera-ready version.
> We truly appreciate the reviewers’ and editor’s time and effort in helping strengthen the clarity and impact of our work.